# Use of GPS to measure external load and estimate the incidence of muscle injuries in men's football: A novel descriptive study

**Marc Guitart[1], Martí Casals[1,2,3]\*, David Casamichana[4], Jordi Cortés[5], Francesc Xavier Valle[1], Alan McCall[6], Francesc Cos[3], Gil Rodas[1,7]**

**1** Performance and Medical Department, FC Barcelona, Sant Joan Despí, Barcelona, Spain, **2** Sport and Physical Activity Studies Centre (CEEAF), Faculty of Medicine, University of Vic—Central University of Catalonia (UVic-UCC), Catalonia, Spain, **3** National Institute of Physical Education of Catalonia (INEFC), University of Barcelona, Barcelona, Spain, **4** Real Sociedad de Fútbol, Donostia-San Sebastián, Spain, **5** Department of Statistics and Operations Research, Universitat Politècnica de Catalunya/BarcelonaTech, Barcelona, Spain, **6** School of Applied Sciences, Edinburgh Napier University, Edinburgh, United Kingdom, **7** Sports Medicine Department, Clinic/Sant Joan de Deu Hospital, Barcelona, Spain

\* marticasals@gmail.com

**Data Availability Statement:** The data are owned by a Spanish sports club that wants to remain anonymous and did not grant permission to make

## Abstract

Measurement of external load in players provides objective information to optimise the weekly balance between training and recovery to improve performance and prevent injuries. Our aim was to evaluate the incidence of sports-related muscle injuries of the lower limb in relation to external load, measured by global positioning system (GPS), in football players. A descriptive study was carried out. Data were collected from 71 professional male football players (30 professionals and 41 youth players) from an elite football club competing in the Spanish and European League in the 2017–2018 season. As external load variables, we measured High Metabolic Load Distance (HMLD), High Speed Running (HSR), Player Load (PL), and Total Distance (TD) through GPS. Injury rate (IR) was calculated both in relation to such GPS load metrics and to load exposure time. We considered categories (youth and professional), playing positions (centre back, full back, midfielder, and forward), and training day with respect to match-day (-4MD, -3MD, -2MD, -1MD, MD, +1MD, +2MD). The GPS load metrics HMLD, HSR, PL, and TD showed very similar patterns across categories and positions, but varied according to training session or MD. The highest loads were observed on MD and three days prior to the match (-3MD). Similarly, the overall IR, both calculated per load exposure time and per GPS load metrics, was highest on MD and -3MD. Again, no differences were observed between youth and professional players. Midfielders demonstrated the highest IR in all metrics, followed by the forwards. In conclusion, this study suggests that external load and incidence of muscle injuries are directly proportional. Therefore, the measurement of more external load variables other than load exposure time, such as the GPS metrics HMLD, HSR, PL, and TD may help to describe the pattern and magnitude of injuries. Future studies based on ours may help to further improve the understanding of the incidence of muscle injuries on the basis of external loads measurements in different football teams.

the original data publicly available. The club has the
right to choose which information, results and data
can be made public and has granted the access to
these data to the authors only for research aims.
For data requests please contact the Data
Protection Officer of FC BARCELONA through the
email address dpo@fcbarcelona.cat.

**Funding:** JC and MC acknowledge the financial
support from Ministerio de Ciencia e Innovación y
Ministerio de Universidades (Ref: PID2019-
104830RB-I00 / AEI DOI: 10.13039/
501100011033).

**Competing interests:** The authors have declared
that no competing interests exist.

## Introduction

Muscle injuries represent the most common and highest injury burden among professional
and youth male footballers [1]. Training sessions aligned with game demands might increase
the risk of such injuries. Consequently, strength and conditioning coaches and medical teams
must focus on designing adequate training programmes to balance the risk of injury with the
benefits of preparing players optimally.

Studies on football injuries typically report injury rates (IR) as the number of injuries per
1000 player-hours [2, 3] either among professional players [4, 5], or youth footballers [6, 7].
However, it may be more important to understand the injury incidence considering other load
variables rather than just 'time'. Indeed, measures of IR based on load exposure time may be
hard to interpret because they are highly dependent on the sport and context. On the contrary,
other load variables, such as the Global Positioning System (GPS) metrics high metabolic load
distance (HMLD), high-speed running (HSR), player load (PL), and total distance (TD) may
better describe injury incidence.

Football teams commonly take advantage of GPS data to take decisions on drills within ses-
sions [8, 9]. However, to date, we lack reliable data on IR according to GPS metrics during a cer-
tain period of time in order to describe the real pattern of the load and the rates of injuries in
each team [10]. Descriptive analysis helps to understand potentially relevant problems and aims
to generate new hypotheses or ideas for subsequent studies. Moreover, few studies provide
information on youth and senior players training and playing with the same methodology.

The aim of our study was to reliably evaluate the incidence of sports-related muscle injuries
of the lower limb both according to the exposure time and to the GPS metrics HMLD, HSR,
PL, and TD, considering category, position, and day with respect to the match, during a typical
season. To our knowledge, it is the first time that such analysis is performed in elite footballers.

## Methodology

### Study design, setting, and participants

A descriptive study was carried out in 71 players from an elite football club competing in the
Spanish and European Champions League during the 2017–2018 season.

Forty-one players were from youth categories: 22 from the under 18 (U-18) team (mean
age ± SD = 16.01 ± 0.71 years, weight = 67.1 ± 8.72 kg, and height = 176.3± 8.11 cm); and 19
from the U-19 team (mean age ± SD = 17.02± 0.70 years, weight = 69.1± 7.34 kg, and
height = 177.7 ± 8.11 cm). Thirty players were from the professional senior category FC Barce-
lona B, that plays in the 2$^{nd}$ Spanish division (mean age ± SD = 20.5 ± 5.45 years, weight = 74.0
± 5.45 kg, height = 179.9 ± 7.64 cm).

### Description of FCB structured microcycle training model during season

Data were collected from the U-18 team (45 weeks and 206 sessions of training, and 54 official
matches); the U-19 team (44 weeks and 203 sessions of training, and 55 official matches, of
which 45 at the national level and 10 at the international level); and the professional male team
(46 weeks and 222 sessions of training, and 45 official matches). International duty matches
and training were excluded. We analysed all training weeks, regardless of their structure or the
number of days between matches. GPS load metrics were analysed with respect to the number
of days before (-MD) or after (+MD) a match, respectively for loading and recovery sessions.
The + 1MD was the session held the day after match day, in which the players were divided
into two groups: those who had played for >60 minutes during the match (they performed
recovery tasks i.e., rondo, mobility, or low intensity runs); and those who had played for <60

minutes or had not participated (they performed compensatory load tasks, i.e., strength circuits, reduced matches, or high intensity runs). These two groups were combined for the analysis. Specific targets for each training day can be found in S1 Appendix. The external loads of all preseason training sessions and preseason friendly games were categorized as preseason.

### Collection and analysis of GPS load metrics

GPS data were collected using the WIMU PROTM device (RealtrackSystems S.L., Almeria, Spain). Intra- and inter-unit reliability was acceptable (intra-class correlation coefficient value was 0.65 for the x-coordinate, and 0.85 for the y-coordinate) for the systems analysed [11]. The data collected were analysed using the SPROTM Software (version 927; RealtrackSystems, Almeria Spain), which exports the data in RAW format.

### Definition of the external load variables

The external load for training and match has been studied considering the following volume variables: total time (TT; min); high metabolic load distance (HMLD, m->25.5W·kg$^{-1}$-distance covered above 21 km·h$^{-1}$, which is the threshold defined for this study); high speed running (HSR; m speeds above 21 km·h$^{-1}$); player load (PL, arbitrary units); total distance (TD; mt.). Intensity variables of external load were also calculated: HMLD·min; HSR·min; PL·min; and TD·min. Averages for all variables (volume and intensity) were calculated as the mean for all sessions [12]. They were then stratified for each category (youth/professional); playing position (centre-back, full-back, midfielder, forward); and training/match day (-4MD, -3MD, -2MD, -1MD, MD, +1MD, +2MD, Preseason). Variables are defined in more detail in Table 1 [13, 14].

### Muscle injury recording

This study followed consensus guidelines on the definitions and data collection procedures for football injury studies described by UEFA [15]. Injuries had to occur during training or match. If they caused the player to be absent from at least the following training session or match, they were classified as time-loss (TL) injuries. On the contrary, no time-loss (NTL) injuries were defined as injuries requiring medical assessment (MA) but not causing

**Table 1. Definition of external load variables.**

| VARIABLE | DEFINITION | INJURY INCIDENCE | |
|---|---|---|---|
| **Total time (TT)** | Total Time (hours) from the start to the end, pauses included. | $IR_{TT} = N/TT$ ($10^{-3}$h) | Injury rate with respect to time |
| **Total Distance (TD)** | Total distance covered (m): this includes walking, jogging, HSR, and sprinting. | $IR_{TD} = N/TD$ ($10^{-7}$m) | Injury rate with respect to external load variables measured through GPS |
| **High Metabolic Load Distance (HMLD)** | Distance covered (m) by a player when his Metabolic Power is above 25.5W/kg (corresponding to running at a constant speed of 5.5m/s, or to significant acceleration and deceleration activities). | $IR_{HMLD} = N/HMLD$ ($10^{-5}$m) | |
| **High-speed running distance (HSR)** | Total distance covered (m) at a speed of >21km/h. | $IR_{HSR} = N/HSR$ ($10^{-5}$m) | |
| **Player Load (PL)***  | Load Index (arbitrary units; AU) computed using the following formulas: $$PL_n = \sqrt{\frac{(X_n - X_{n-1})^2 + (Y_n - Y_{n-1})^2 + (Z_n Z_{n-1})^2}{100}}$$ $$PLaccumulated = \sum_{n=0}^{m} PL_n x 0.01$$ n is the order index over time; $t_n$ is the present time; $PL_n$ is the Player Load calculated at $t_n$ (Instant Player Load); $X_n$, $Y_n$, and $Z_n$ are the values of BodyX, BodyY and BodyZ at $t_n$. | $IR_{PL} = N/PL$ ($10^{-5}$ AU) | |

Notes: $IR_{TT}$ = Injury Rate per total time; $IR_{TD}$ = Injury Rate per Distance; $IR_{HMLD}$ = Injury Rate per HMLD; $IR_{HSR}$ = Injury Rate per HSR; $IR_{PL}$ = Injury Rate per PL; N = number of injuries.

withdrawal from training or matches. Injuries were classified using the OSICS_10 coding system (Orchard Sports Injury Classification System), and muscle injuries were indicated by the code (code OSICS -M--). In this study, we specifically analysed muscle injuries to the lower limb. All the diagnoses were made by the same team doctor. A recurring injury was defined as an injury of the same type occurring at the same anatomical location as a previous lesion within two months after return to play [6, 7]. To calculate injury severity [16], we based on a UEFA proposal [15, 17, 18] from the first week of the preseason. Severity was determined according to the number of days from injury occurrence until the end of medical leave, ranging from mild (1–7 days), to moderate (8–28 days), and severe (>28 days).

## Statistical analysis

Absolute (n) and relative (%) frequencies for categorical variables, and measures of central tendency and dispersion for continuous variables were calculated.

Average of external load variables (volume and intensity) were calculated as the mean over all season. Global and stratified averages by category (youth/professional), position (centre back, full back, midfielder, forward), and matchday (-4MD, -3MD, -2MD, -1MD, MD, +1MD, +2MD, Preseason) are presented. The Box plots shown contain markers for the median of the data and a box indicating the interquartile range (IQR).

To calculate the injury incidence, we computed the load variables and subdivided the exposure into game exposure (GE) and training exposure (TE). These data were recorded by a member of the technical staff. We calculated $IR_{TT}$ as the number of injuries per 1000 hours ($10^3$h) of player-hours of exposure; $IR_{HMLD}$ as the number of injuries per 100,000 player-meters ($10^5$m) of HMLD; $IR_{HSR}$ as the number of injuries per $10^5$ player-meters ($10^5$m) of HSR; $IR_{PL}$ as the number of injuries per $10^5$ arbitrary units ($10^5$AU) of PL; and $IR_{TD}$ as the number of injuries per $10^7$ player-meters ($10^7$m) of TD. All incidences were also computed separately for TL and NTL injuries, and according to the different levels of severity.

All analyses were performed using the R statistical package (The R Foundation for Statistical Computing), version 3.6.0, specifically the *data.table* and *epitools* (https://cran.rproject.org/web/packages/epitools/index.html) packages.

## Ethics approval

Players provided written informed consent to participate. We excluded players who had not started the competitive period of the season with their teams. The study protocol was approved by the FCB Medical Committee and the local research ethics committee on Science and Ethics of the Barça Innovation Hub (Football Club Barcelona; n° 2019FCB28). This study conformed to the recommendations of the Declaration of Helsinki.

## Results

### External load

The total analysis included 631 training sessions and 154 official matches, for a total of 12,340 hours of training and 2,077 hours of match play.

Fig 1 shows GPS load metrics measurements: HMDL (m), HSR (m), PL (AU), and TD (m) for each category, position, and training/match day.

The global median (IQR) of HMLD, HSR, PL, and TD were 641.86m (408.47–975.60), 122.53m (49.55–270.56), 58.80AU (41.10–80.50), and 4,328m (3,174–5,923), respectively.

All four parameters showed very similar distributions according to position and category, and they varied according to training/match day, with the highest loads observed on match

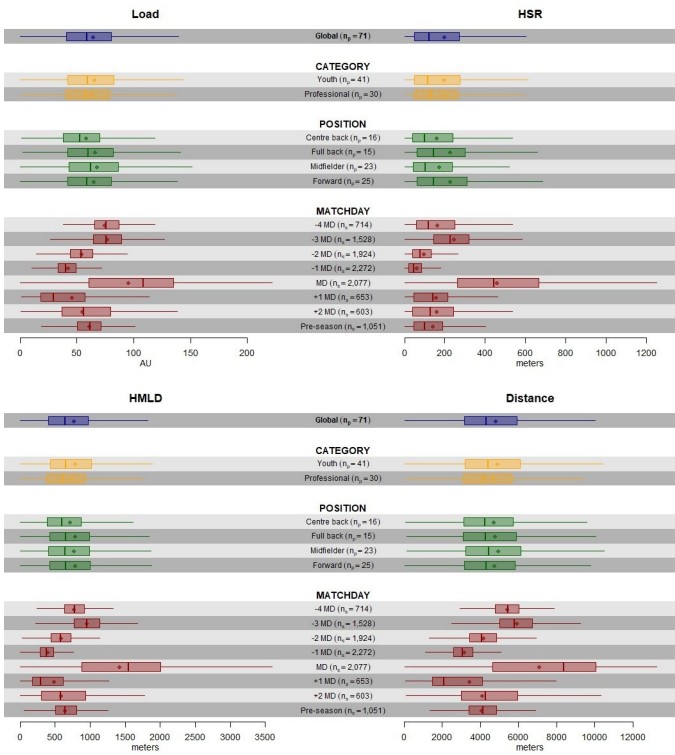

**Fig 1. Boxplots without outliers of GPS load metrics by category, position, and training/match day.** HMLD is high metabolic load distance; HSR is high speed running; $n_p$ is number of players; and $n_s$ is number of sessions. Note: Distance refers to Total Distance; Load refers to Player Load.

day (MD) and three days prior to the match (-3MD). These differences appear to be more pronounced for HSR (median of MD = 444.41 metres and median of -3MD = 227.63 metres), in comparison to the other variables.

In Fig 2, we show the variables for external load intensity: HMLD/min, HSR/min, PL/min, and TD/min.

The global median (IQR) of HMLD, HSR, PL, and TD intensities were 8.94 m/min (6.39–13.31), 1.75 m/min (0.72–3.94), 0.80 AU/min (0.70–1.00), and 59.7 m/min (50.60–75.20), respectively.

These parameters also evinced very alike distributions according to position and category, and oscillated according to training/match day, with the most demanding loads again observed on match day (MD) and three days prior to the match (-3MD). In this case, the difference of MD and -3MD with the other days is high for all intensity parameters.

### Frequency of muscle injuries

During the entire season, we registered a total of 34 episodes of muscle injuries, of which 19 were TL injuries (Table 2).

### Incidence of muscle injuries

**Muscle injury incidence per exposure hours.** The overall incidence per exposure hours (Fig 3) was 2.57 injuries/$10^3$h, with the highest values on MD (4.55/$10^3$h), followed by -3MD (4.07/$10^3$h).

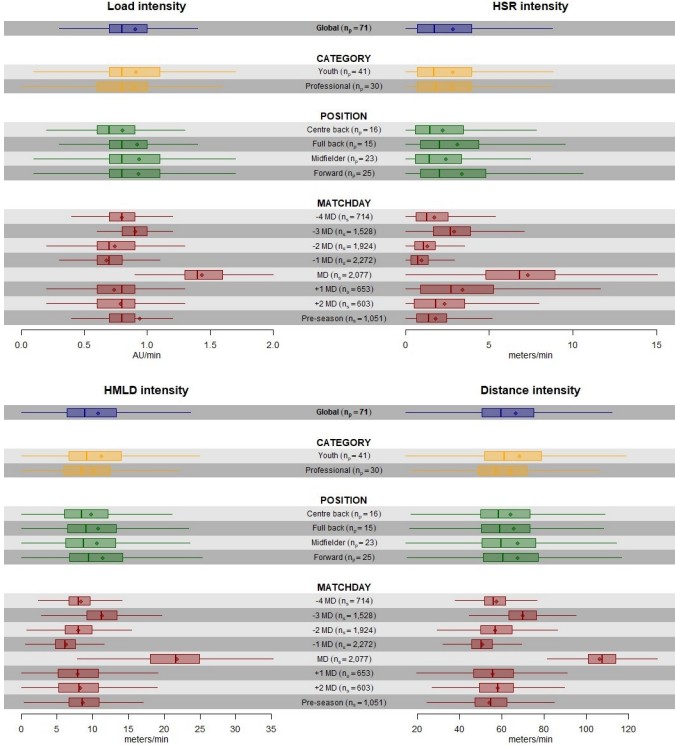

**Fig 2. Boxplots without outliers of GPS load metrics HMLD, HSR, PL, and TD per minute for each category, position, and training/match day.** $n_p$ = Number of players; $n_s$ = Number of sessions.

**Table 2. Muscle injury type.**

|  | Time loss (TL) | No time loss (NTL) | Total |
|---|---|---|---|
| **CATEGORY** |  |  |  |
| **Youth** | 11 | 11 | 22 |
| **Professional** | 8 | 4 | 12 |
| **TOTAL** | 19 | 15 | 34 |
| **POSITION** |  |  |  |
| **Centre Back** | 2 | 3 | 5 |
| **Full back** | 1 | 2 | 3 |
| **Midfielders** | 9 | 8 | 17 |
| **Forwards** | 7 | 2 | 9 |
| **TOTAL** | 19 | 15 | 34 |
| **MATCH DAY** |  |  |  |
| **-4MD** | 0 | 2 | 2 |
| **-3MD** | 6 | 3 | 9 |
| **-2MD** | 1 | 4 | 5 |
| **-1MD** | 1 | 3 | 4 |
| **MD** | 9 | 2 | 11 |
| **Pre-season** | 2 | 1 | 3 |
| **TOTAL** | 19 | 15 | 34 |

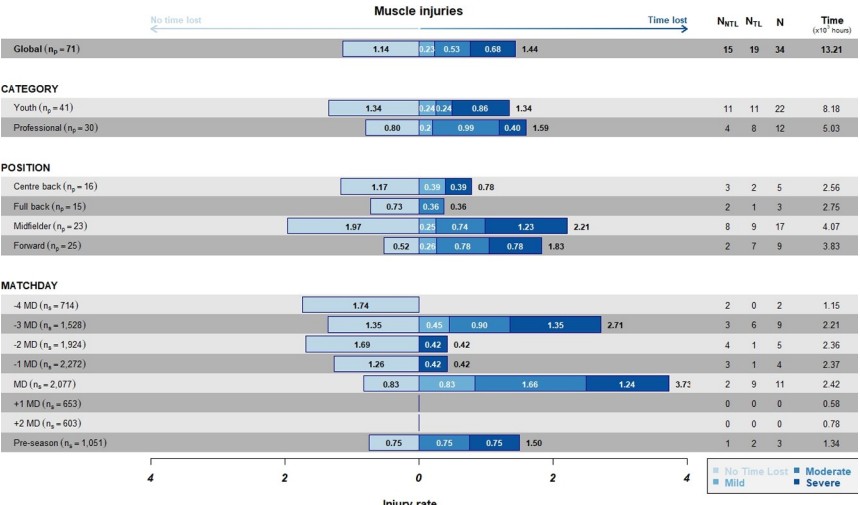

**Fig 3. Incidence of TL and NTL injuries per 10³h of exposure stratified by category, position, and days to/from the match.** Numbers inside bars represents the incidence according to the severity.

The incidence of TL injuries was higher on MD ($3.71 \times 10^3$h), followed by -3MD ($2.71 \times 10^3$h). As for categories, the highest values were reported in professionals ($1.59 \times 10^3$h), in comparison to youth ($1.34 \times 10^3$h). Finally, as for positions, the highest rate was observed among midfielders ($2.21 \times 10^3$h), followed by forwards ($1.82 \times 10^3$h).

**Muscle injury incidence per HMLD.** The overall incidence of muscle injuries per $10^5$m of HMLD (Fig 4) was similar to the incidence per hours of exposure: the highest incidence was observed on -3MD ($0.41 \times 10^5$m), followed by MD ($0.29 \times 10^5$m) and preseason. Moreover, overall incidence of muscle injuries per $10^5$m of HMLD was higher in the professional category ($0.25 \times 10^5$m) than in the youth ($0.20 \times 10^5$m); and midfielders presented the highest values ($0.35 \times 10^5$m), followed by forwards ($0.27 \times 10^5$m).

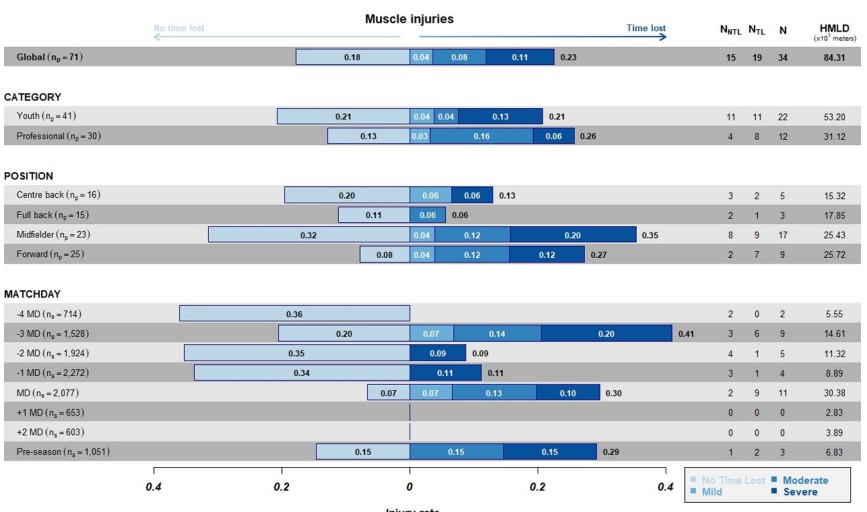

**Fig 4. Incidence of TL and NTL injuries per $10^5$m of high metabolic load distance stratified by category, position, and days to/from the match.** Numbers inside bars represent the incidence according to the severity.

**Muscle injury incidence per HSR.** The overall muscle injury incidence per $10^5$m of HSR (Fig 5) was the highest on -3MD ($1.59 \times 10^5$m), followed by the preseason ($1.36 \times 10^5$m); in the professional category ($0.94 \times 10^5$m); and among midfielders ($1.60 \times 10^5$m) (Fig 5).

**Muscle injury incidence per PL.** The overall muscle injury incidence per $10^5$a.u. of PL (Fig 6) was the highest on -3MD ($5.06 \times 10^5$a.u.), followed by the match day ($4.39 \times 10^5$a.u.); in the professional category ($3.03 \times 10^5$a.u.); and among midfielders ($4.03 \times 10^5$a.u.).

**Muscle injury incidence per TD.** The overall muscle injury incidence per $10^7$m of TD (Fig 7) was higher on -3MD ($6.61 \times 10^7$m), followed by the match day ($5.91 \times 10^7$m); in the professional category ($4.10 \times 10^7$m); and among midfielders ($5.52 \times 10^7$m).

## Discussion

Our study showed that MD and -3MD were the days with both the highest external load, and the highest injury incidence either calculated per exposure time or per all the GPS load metrics (HMLD, HSR, PL, and TD). Also, professional players and midfielders displayed the highest injury rate for all the GPS metrics, but not for exposure time.

### External loads

**Differences in GPS load metrics between professional and youth players.** In agreement with previous studies, there were no differences on the GPS load metrics between youth and professionals [19]. In fact, as previously shown, HMLD, HSR, PL, and TD were just slightly higher in youth, in comparison to the professional category (Figs 1 and 2) [20]. In our study, a common training methodology was shared among the three teams involved, using very similar training exercises and the same weekly load structure, and this can explain such lack of differences between categories. It is important to consider this comparison since it is common for youth players to be called up to train with the senior professional team. Therefore, from a performance and injury perspective, it would be important for these players to be prepared and able to cope with the volumes and intensities of the senior team.

**Differences in GPS load metrics according to playing position.** Our findings showed similar external GPS load metrics according to playing position, even if full backs and forwards

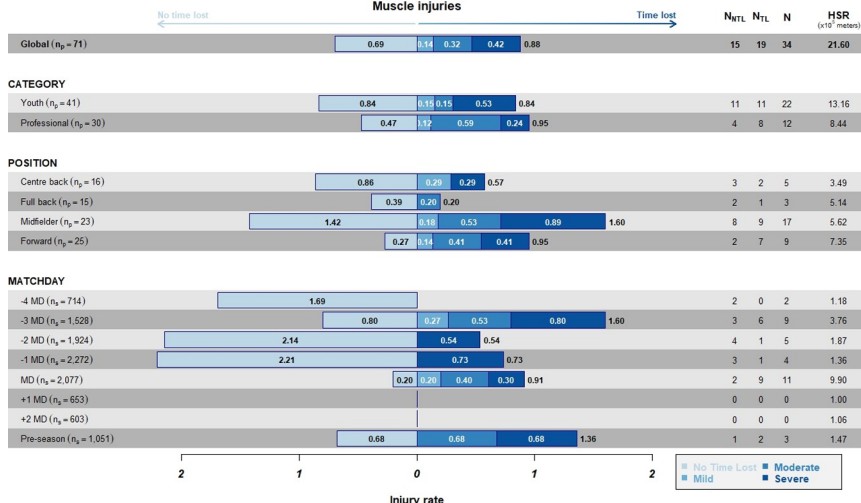

**Fig 5. Incidence of TL and NTL injuries per $10^5$m of high-speed running stratified by category, position, and days to/from the match.** Numbers inside bars represents the incidence according to the severity.

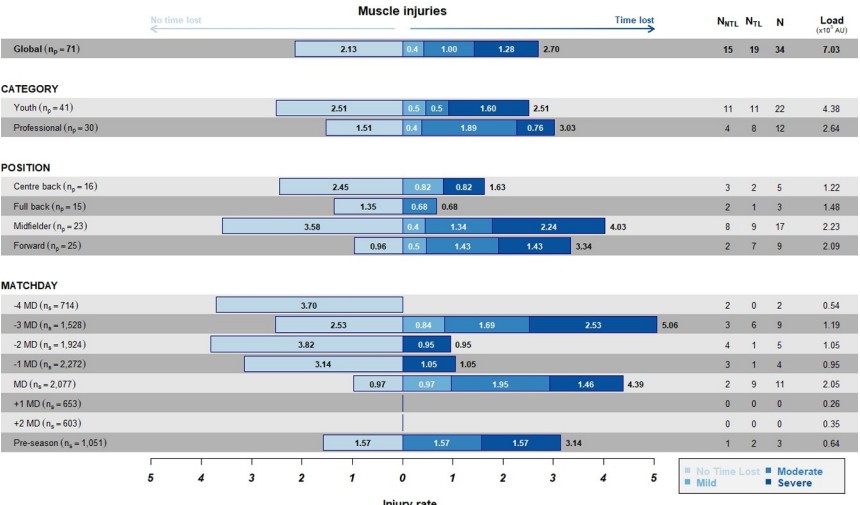

**Fig 6. Incidence of TL and NTL injuries per $10^5$au of PL stratified by category, position, and days to/from the match.** Numbers inside bars represents the incidence according to the severity.

showed slightly higher HSR volume (m) and intensity (m*min). On the contrary, past research demonstrated that defenders cover less high intensity distances compared to other outfield positions [21–23]. These differences might be explained by tactical roles of full backs and forwards, which are unique to the playing style of each club.

## Incidence of muscle injuries

**Differences between IR per exposure hours and IR per GPS load metrics.** We did not find any differences between $IR_{TT}$ calculated according to the exposure hours and IR calculated according to the GPS load metrics HMDL, HSR, PL, and TD. The information in terms of exposure hours is very general, whereas HMDL, HSR, PL, and TD are more specific for players' physical demands. Although HMDL, HSR, PL, and TD were greater at MD, the IR

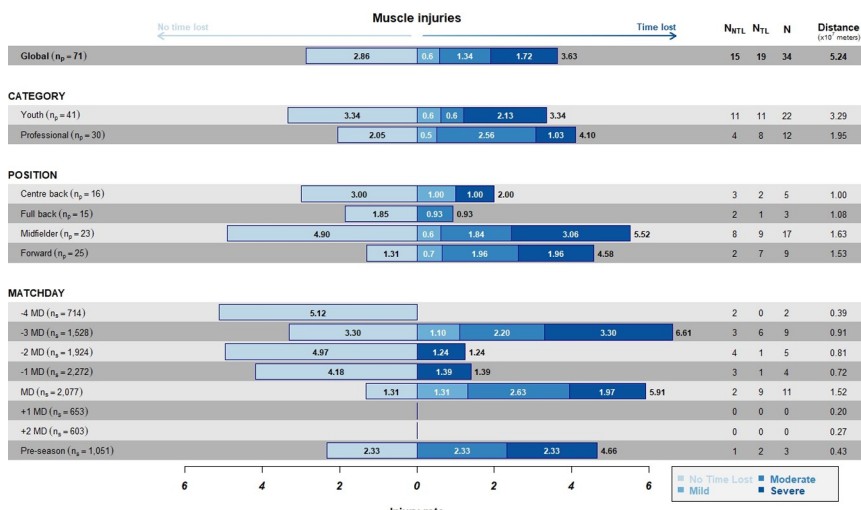

**Fig 7. Incidence of NTL and TL injuries per $10^7$m distance of TD stratified by category, position, and days to/from the match.** Numbers inside bars represents the incidence according to the severity.

calculated according to these GPS metrics was higher at -3MD ($0.41 \times 10^5$m HMLD; $1.60 \times 10^5$m HSR; $5.06 \times 10^5$a.u. PL; $6.61 \times 10^7$m TD). IR calculated according to TT was higher at MD ($3.73 \times 1000$h), followed by -3MD, very comprehensive in the model of the FCB structured microcycle.

**Differences in IR between professionals and youth players.** In our study, we did not see a big difference between categories in IR calculated for any of the external load variables (exposure hours, HMDL, HSR, PL, and TD). However, in agreement with previous studies, all variables were slightly higher among professionals ($IR_{TT} = 1.59$; $IR_{HMLD} = 0.26$; $IR_{HSR} = 0.95$; $IR_{PL} = 3.03$; $IR_{TD} = 4.10$), in comparison to youth players ($IR_{TT} = 1.34$; $IR_{HMLD} = 0.21$; $IR_{HSR} = 0.84$; $IR_{PL} = 2.51$; $IR_{TD} = 3.3$). Our results could be explained by the fact that we analysed three teams from the same club, with the same training methodology and the same game model. In the future, more clubs with training methodologies and game models that are different from ours should be analysed to confirm our findings.

**Differences in IR according to playing position.** By analysing injury incidence according to playing position, we found that midfielders showed the highest IR for all metrics studied, followed by forwards, with a notable difference with centre-backs and full-backs. Another study by Bacon et al. [24], provided positional incidence rates per 1000 h. The authors found that central midfielders displayed the highest risk of injury ($14.22 \pm 15.46$ per 1000 h), and lateral midfielders had the lowest risk of injury ($2.15 \pm 2.49$ per 1000 h). We postulate that, in general, there are no differences between categories of players in our club: the training is similar, and it is common for players to train and compete in teams in different groups of age during a season.

## Limitations

The most important limitation of this study is the small sample (71 players). Moreover, only one club was involved, and just a season was analysed. However, we included 631 training sessions and 154 official matches, for a total of 12,340 hours of training and 2,077 hours of match play. Similar future studies using external load variables other than time must be carried out in more football clubs, with different training methodologies, practices, and game models. These studies will allow to better understand the influence of specific training methodologies on injury incidence based on external load exposure.

## Practical applications

Our data could help youth and professional teams to better understand the patterns of external loads (such as exposure time, PL, TD, HMLD, and HSR), and their incidence on muscle injuries, in different categories, positions, and days. Finally, these findings could be used to make daily management decisions, such as: i) share players between different categories in the same club; ii) exchange demarcation players within the same game model, interpreting how they could modify their physical behaviour; and iii) manage the tasks in the training sessions.

## Conclusions

This is the first study that reliably describes the incidence of muscle injuries in male professional and youth players according to GPS metrics. We show that the IR was higher on MD when calculated per hours, and on -3MD when calculated per HMLD, PL, TD and HSR. Midfielders demonstrated the highest IR in all metrics, followed by the forwards. Finally, to confirm our findings, future studies must incorporate more football players, both from youth and professional football, and include several seasons in the analysis.

## Supporting information

**S1 Appendix. Specific targets for each training day relative to the match.**
(DOCX)

## Acknowledgments

The authors would like to thank the players of the Football Club Barcelona for their participation in this study.

## Author Contributions

**Conceptualization:** Marc Guitart, Martí Casals, Francesc Xavier Valle, Francesc Cos, Gil Rodas.

**Formal analysis:** Martí Casals, Jordi Cortés.

**Methodology:** Marc Guitart, Martí Casals, Jordi Cortés, Alan McCall, Gil Rodas.

**Project administration:** Martí Casals.

**Supervision:** Marc Guitart, Martí Casals, Jordi Cortés, Gil Rodas.

**Validation:** Martí Casals.

**Visualization:** Martí Casals, Jordi Cortés.

**Writing – original draft:** Marc Guitart, Martí Casals, David Casamichana, Jordi Cortés, Francesc Xavier Valle, Alan McCall, Francesc Cos, Gil Rodas.

**Writing – review & editing:** Marc Guitart, Martí Casals, David Casamichana, Jordi Cortés, Francesc Xavier Valle, Alan McCall, Francesc Cos, Gil Rodas.

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
