## [Decision Letter · Decision Letter 0]

11 Nov 2020

PONE-D-20-22817

Muscle injury incidence according to exposure of time and global positioning satellite system metrics: A descriptive study from a senior men’s and academy team belonging to a Spanish professional club

PLOS ONE

Dear Dr. Casals,

Thank you for submitting your manuscript to PLOS ONE. After careful consideration, we feel that it has merit but does not fully meet PLOS ONE’s publication criteria as it currently stands. Therefore, we invite you to submit a revised version of the manuscript that addresses the points raised during the review process.

We look forward to receiving your revised manuscript.

Kind regards,

Fabrizio Perroni

Academic Editor

PLOS ONE

Journal Requirements:

2. Thank you for stating the following in the Competing Interests Section of your manuscript:

"No funding was received for the present investigation."

 "No."

5. Please include your tables as part of your main manuscript and remove the individual files. Please note that supplementary tables (should remain/ be uploaded) as separate "supporting information" files.

Reviewers' comments:

Reviewer's Responses to Questions

**Comments to the Author**

1. Is the manuscript technically sound, and do the data support the conclusions?

Reviewer #1: No

Reviewer #2: Yes

2. Has the statistical analysis been performed appropriately and rigorously? 

Reviewer #1: I Don't Know

Reviewer #2: No

3. Have the authors made all data underlying the findings in their manuscript fully available?

Reviewer #1: Yes

Reviewer #2: Yes

4. Is the manuscript presented in an intelligible fashion and written in standard English?

Reviewer #1: No

Reviewer #2: No

5. Review Comments to the Author

Reviewer #1: Thank you for the opportunity to review the manuscript titled ‘Muscle injury according to exposure of time and global positioning satellite system metrics: A descriptive study from a senior men’s and academy team belonging to a Spanish professional club’. Clearly there has been substantial data collection and analysis for this paper.

General comments:

Throughout the manuscript there are formatting issues for example line 63 to 69 does not constitute a paragraph, similarly lines 71 to 73, 155 to 156, 242 to 246, 269 to 271, 290 to 298, 429 to 432,452 to 454, 503 to 513, 517 to 534 need to be formatted correctly into paragraphs. Unfortunately, this gives an untidy appearance to the reader. Most of the authors of this study I would assume their first language is not English. While I do empathise that writing in a non-native language is difficult for this study to reach publication it would need extensive re-writing and structuring.

Introduction would be better if this section provided a clear argument for the need to calculate injury incidence using GPS metrics. At the moment this is missing in my view. There little argument and this section is very descriptive. In general, the introduction is too long and needs work on the content and structure. For example, if the main aim of the paper is to use GPS metrics to calculate injury incidence then injury incidence (specifically muscle injury) and the limitations of how they are currently calculated should be addressed in details. Line 91 to 102 focuses solely on GPS might be better to alter the emphasis and provide an argument for calculating injury incidence using these metrics, so practitioners understand the risk associated with specific training aims i.e., greater distances covered increases muscle injury risk. Check formatting throughout 2-4 sentences does not justify a new paragraph. Overall this section needs to be re-wrote and structured.

Discussion, the title does not reflect the aims of the main content i.e., muscle injury according to GPS metrics is not discussed until the fourth section of the discussion. Again, the discussion is too long and would benefit from more concise writing. More detailed and structured discussion of your findings is required. The first finding that is discussed ‘all parameters of player external load (HMLD,HSR, PL and TD) in volume and intensity are according the club structure plan of training-match day’; suggesting this is the major/novel finding from your data. Surely the major/novel finding should be muscle injury incidence according to GPS metrics? The fact that volume and intensities are inline with the club training plan might offer some explanation as to why muscle injury incidence according to GPS metric did not differ between senior and academy players. Overall this section needs to be re-wrote and structured for example, Line 495: ‘most important finding’…it has not been discussed until page 4 of the discussion.

Reviewer #2: MAJOR COMMENTS

• There are many grammar and typing errors throughout the manuscript. The manuscript should be revised thoroughly. There are many English mistakes. In my opinion, there are mistakes from the beginning, also in the title for instance …” to exposure of time”…

Examples of typing errors:

Line 150 ….” Throght-out”

Line 17: “ fo Cataonia”

Line 39: “ 30 professional male and 41 male youth academy”

Also check verbs: sometimes present other past tenses… (ie line 319)

Line 404: “According planning the training week”?

Line 453-454

Line 509-510. Difficult to understand the meaning of the numbers in brackets.

[there are too many mistakes and errors for a reviewer to correct them all, please check carefully the manuscript. It is not acceptable as it is now]

• The study was approved by the Ethics Committee of the football club. I wonder if this is enough for the standards of the journal. Usually an independent Ethics Committee of a Hospital or University should approve the studies.

• In the abstract, the authors say in line 37 that this is a descriptive epidemiological study, however I think that this study is descriptive; but it is not epidemiological. Consequently, this word should be replaced in the key words list. The same applies for line 134.

• The methods section should be improved, because there are some crucial issues. It is unclear the term HMLD (High Metabolic Load Distance). On the one hand, it is difficult to understand the meaning of the term. Secondly, the authors should explain what the metabolic power 25.5 W/kg comes from. The reference number 12 does not seem to explain it. Also, the authors state that the HMLD is the distance covered above 21 km/h, however they do not explain the reason for that. Moreover, it is questionable that 21km/h is high speed running for all players, this would suggest that all players have the same running speed. The authors should explain the reason for choosing 21 km/h instead of 20 km/h or other velocity.

• How were accelerations and decelerations defined?

• Last paragraph in page 9 (Lines 199-204) is difficult to understand because of the errors; besides, it seems to be repeated in lines 211-216.

• The methodology should be clearer about the registered days. Did the authors separate season and preseason training days? If so, it should clearer. Moreover, they should also clarify if footballers played any matches during the preseason.

• Line 159-160. The authors state: “ We analysed all weeks of training, regardless of their structure or the number of days between matches”. It is quite usual that clubs organize matches during the week-end but also during the week, particularly friendly matches. How did the authors deal with these mid-week matches and the registration of the training sessions?

• Line 243-244: difficult to understand. It is probably not well written. In any case, the authors state that “…we used during the first week of the season” which is difficult to understand.

• An statistical analysis should be performed in order to compare load and injuries amongst groups ( youth vs senior, positions etc). In this line, sometimes authors assume that two values are different, but because there is no analysis it is not possible to know if the difference is significant. For example, line 335: is 1.59 larger than 1.34? How large is this difference?

• Line 297. It would appropriate to mention and explain “physical intensity variables” in the methods section. Also, check for the English of this terminology.

• The discussion section is poor. Authors should discuss their results and explain them, giving reasons for it. In addition, they should compare to other studies.

• Practical applications: This paragraph should be rewritten in order to better clarify the real application of this study.

• The quality of the graphs is poor. It is very difficult to read the numbers and the letters

MINOR COMMENTS

• Line 38: All the participants were not professional players, and therefore it should be changed.

• Line 162. The authors wrote …” began their competitive period for the season in question..”. What does this mean?

• Line 144. It is difficult to understand the reason for writing that the players conducted pre-participation evaluation.

• 150 line: it is not correct

• There are many errors throughout the manuscript

• Check References: there are many mistakes, for example in lines 597, 608, 623, 669…

6. PLOS authors have the option to publish the peer review history of their article (what does this mean?). If published, this will include your full peer review and any attached files.

Reviewer #1: No

Reviewer #2: No

---

## [Author Response · Author response to Decision Letter 0]

20 Dec 2020

Dear Editor,

We would like to thank you for giving us the opportunity of submitting a revision of the manuscript entitled “Descriptive analysis of GPS as exposure measures and its use in estimating the muscle injury incidence from a senior men’s and academy football team of a Spanish professional club”.

We have carefully read the comments and suggestions that have certainly improved the manuscript. Additionally, a list of changes and answers to the reviewers’ comments has been submitted.

Sincerely,

Corresponding author: Martí Casals. Sport and Physical Activity Studies Centre (CEEAF), University of Vic – Central University of Catalonia (UVic-UCC). Address: C. Dr. Antoni Vilà Cañellas, s/n, 08500 Vic, Spain. Tel: +34 647053785. 

E-mail: marticasals@gmail.com, marti.casals1@umedicina.cat

Thank you for the opportunity to review the manuscript titled ‘Muscle injury according to exposure of time and global positioning satellite system metrics: A descriptive study from a senior men’s and academy team belonging to a Spanish professional club’. Clearly there has been substantial data collection and analysis for this paper. 

Comments by Authors: We appreciate your positive words.

General comments:

Throughout the manuscript there are formatting issues for example line 63 to 69 does not constitute a paragraph, similarly lines 71 to 73, 155 to 156, 242 to 246, 269 to 271, 290 to 298, 429 to 432,452 to 454, 503 to 513, 517 to 534 need to be formatted correctly into paragraphs. Unfortunately, this gives an untidy appearance to the reader. Most of the authors of this study I would assume their first language is not English. While I do empathise that writing in a non-native language is difficult for this study to reach publication it would need extensive re-writing and structuring. 

Reply: Thanks for your comments. Following the suggestion, we have improved most of these paragraphs. 

Introduction would be better if this section provided a clear argument for the need to calculate injury incidence using GPS metrics. At the moment this is missing in my view. There little argument and this section is very descriptive. In general, the introduction is too long and needs work on the content and structure. For example, if the main aim of the paper is to use GPS metrics to calculate injury incidence then injury incidence (specifically muscle injury) and the limitations of how they are currently calculated should be addressed in details. Line 91 to 102 focuses solely on GPS might be better to alter the emphasis and provide an argument for calculating injury incidence using these metrics, so practitioners understand the risk associated with specific training aims i.e., greater distances covered increases muscle injury risk. Check formatting throughout 2-4 sentences does not justify a new paragraph. Overall this section needs to be re-wrote and structured. 

Reply: Thanks for your comments. Following the suggestion, we have improved and moved the following paragraph:

“Football epidemiological studies typically use the injury rate (IR) defined as the number of injuries per 1000 player-hours(2,4–6) to measure the injury incidence (including muscle injury). “

Moreover, we have explained better the main aim of our study. See below:

“Injury incidence is typically calculated by exposure time (match, training, total). However, in sports medicine, some researchers and clinical staff can be also interested in the injury incidence considering other measures of exposure. While the measures based on exposure times, such as the IR, are hard to interpret because they are highly dependent on the sport and context, other exposure types based on GPS metrics are more appropriate to compute injury incidence more homogeneous throughout different disciplines.

Exposures such as the high-speed running (HSR), player load, total distance and high metabolic load distance (HMLD) variables may be useful to describe the injury incidence according to the activity performed. From our knowledge, such descriptive analyses using these exposure parameters have never been performed in elite footballers. 

At the same time it seems important to know the behavior of this training /match external load and injury rate according to training day in respect match play, category and playing positions, due to the different conditional demands that are generated in each one of them, observing important differences in some variables between positions of the same game model [19].”

We also have justified the objectives of our study:

“Our aims were on one hand, to provide an overview of these GPS-based measures through a descriptive analysis and subsequently to study injury incidence behavior grounded in these measures.

For this reason, the specific objectives of this study were: 1) To describe the external load and intensities in players of a academy and professional football players according to playing position and training day respect to match day; and 2) to evaluate incidence rates of sports-related lower limb muscle injuries according to exposure time, HMLD exposure and HSR exposure taking into account the training day respect category, positions and match day during the season. “

Discussion, the title does not reflect the aims of the main content i.e., muscle injury according to GPS metrics is not discussed until the fourth section of the discussion. Again, the discussion is too long and would benefit from more concise writing. More detailed and structured discussion of your findings is required. The first finding that is discussed ‘all parameters of player external load (HMLD,HSR, PL and TD) in volume and intensity are according the club structure plan of training-match day’; suggesting this is the major/novel finding from your data. Surely the major/novel finding should be muscle injury incidence according to GPS metrics? The fact that volume and intensities are inline with the club training plan might offer some explanation as to why muscle injury incidence according to GPS metric did not differ between senior and academy players. Overall this section needs to be re-wrote and structured for example, Line 495: ‘most important finding’…it has not been discussed until page 4 of the discussion. 

Reply: Thanks for your comments. We have updated and improved the Discussion section following your suggestions. 

The title suggests the main aim of the study was to provide injury exposure using GPS metrics however, this is the second aim of the study according to line 124-128. 

Reply: We agree, and we have clarified the aims and we have changed the title following your suggestions. The new title is: “Descriptive analysis of GPS as exposure measures and its use in estimating the muscle injury incidence from a senior men’s and academy football team of a Spanish professional club.“

Abstract

Line 28 to 30: the phrase ‘to guide’ used twice within a single sentence.

Reply: Thank you for your input. We have updated it.

Line 31 to 36: If the main aim of the paper is to describe external loads across one club this should be reflected in the title? If injury according GPS metrics is the main aim please re-order. 

Reply: Thanks. We agree and we have changed the title.

Line 34: muscle injuries – are these lower limbs? Might be better to specific here and also throughout the manuscript.

Reply: Thank you for your input. We have specified it in this line and in the Introduction and Methods (Muscle injuries subsection) sections.

Line 37 to 40: Needs to be re-worded.

Reply: Thank you for your input. We have modified it.

Line 38: Details such as age, height, weight and position would be useful for the reader.

Reply: Thanks. This information is already included in the methods section.

Line 45 to 47: GPS metrics need to be ordered the same throughout the manuscript.

Reply: Thank you for your input. We have considered it.

Line 56 to 58: Need to be re-worded. The novel finding here appears to be that higher intensity training day is associated with increased muscle injury. 

Reply: Thank you for your input. We have modified it.

Introduction

Line 66: Economic reference, no peer review paper? 

Reply: Thanks for your comment. 

We have included these new references:

Batten L Football injury index: English Premier League 2019 review. Marsh JLT Specialty 1-22, 2019.

Ekstrand J Keeping your top players on the pitch: the key to football medicine at a professional level. British Journal of Sports Medicine 47: 723-724, 2013.

Eliakim E, Morgulev E, Lidor R, Meckel Y Estimation of injury costs: financial damage of English Premier League teams’ underachievement due to injuries. BMJ Open Sport Exerc Med 6: e000675, 2020.

Hickey J, Shield AJ, Williams MD, Opar DA The financial cost of hamstring strain injuries in the Australian Football League. British Journal of Sports Medicine 48: 729-730, 2014.

Line 71 to 73: Is this meant to be here? Prior to this you describe injury consequences then after these lines you describe muscles injury. Line 71 to 73 disrupts the flow of your argument, and therefore, needs to be linked with pre and post sentences.

Reply: Thanks for your words. We have moved this paragraph to the paragraph prior to the objectives. 

Line 76: ‘athletes?’ should this be footballers?

Reply: Done

Line 76 to 81: This is too long and very hard to follow. Is it not possible to break this into 2 or 3 sentences?

Reply: Done 

Line 76 to 77: Starts sentence ‘With’ and then repeats ‘with the objective’…may be re-word?

Reply: Done 

Line 77 to 81: Too long for a single sentence and too many points. Might be better broken into 3 sentences 1 point per sentence. 

Reply: Done 

Line 80: ‘also and careful and appropriate’…I am unsure what you are trying to say here, please re-word. Might simply be the case of a typo, but the sentence is unclear.

Reply: Done 

Line 88 to 89: Is there need for parentheses at the end of the sentence.

Reply: Done 

Line 91 to 95: Too long for a single sentence and too many points. Might be better broken into 3 sentences 1 point per sentence. 

Reply: Done 

Line 102: Can you support this statement with a reference?

Reply: Thanks for this point. We have deleted this sentence because it does not provide information.

Line 106: ‘artefacts and artificial alterations’ could you provide a specific example for the reader.

Reply: Thanks for your comment. We have changed this paragraph.

“Today , the ‘fashion’ in football is to investigate association between training load and injuries (15,17–19). However, recent work in this area is actually showing that the results may be influenced by artefacts and artificial alterations and therefore, cannot provide any meaningful applications to practice. For example, Enright et. al. (17) suggest that workload data typically used by professional soccer teams may not be able to discriminate between injury type and/or severity, and Impellizzeri et.al. (20) have also shown that, depending on the characteristics of the sample (injury and data distribution), these artefacts can result in associations that can be statistically significant or compatible with increased or decreased injury risk.”

Line 109: “Despite many publications’ could you provide specific examples of these publications for the reader.

Reply: Thanks for your comment. We have included some references with specific examples of the topic of load. 

Nielsen, R. O., Simonsen, N. S., Casals, M., Stamatakis, E., & Mansournia, M. A. (2020). Methods matter and the ‘too much, too soon’theory (part 2): what is the goal of your sports injury research? Are you describing, predicting or drawing a causal inference?.

Nielsen, R. O., Bertelsen, M. L., Møller, M., Hulme, A., Mansournia, M. A., Casals, M., & Parner, E. T. (2020). Methods matter: exploring the ‘too much, too soon’theory, part 1: causal questions in sports injury research. British journal of sports medicine.

Backes, A., Skejø, S. D., Gette, P., Nielsen, R. Ø., Sørensen, H., Morio, C., & Malisoux, L. (2020). Predicting cumulative load during running using field‐based measures. Scandinavian Journal of Medicine & Science in Sports.

Line 114: ‘incidence is typically calculated using..’ can you support this statement with a reference. You already mention this Line 71 to 73, can you see how this affects the flow for the reader? Each sentence should link.

Reply: Thanks for your comment. We have included the following references at the end of this sentence “Injury incidence is typically calculated using exposure time (match, training, global). “: 

Dick R, Agel J, Marshall SW. National Collegiate Athletic Association Injury Surveillance System commentaries: introduction and methods. J Athl Train 2007;42(2):173-82.

Stovitz SD, Shrier I. Injury rates in team sport events: Tackling challenges in assessing exposure time. Br J Sports Med 2012;46(14):960-963.

Line 114: replace global with total?

Reply: Done

Line 119: add ‘in’ after ‘training day’.

Reply: Done

Line 120: To assist the reader it might be worth defining what is meant by category.

Reply: Done

Line 124: This needs to be clearer for example ‘To investigate external loads and intensities of academy and professional footballers according to…’

Reply: Done

Line 124 to 128: What is the main aim of this study is it to use GPS metrics to calculate injury incidence or explore external loads/intensities. This aim should be reflected in the title and also in the order of the aims.

Reply: We agree with your comment. We have changed the title considering the two objectives described previously.

Methods

Line 134 to 146: Start almost all sentences with ‘We’, might be better to selected different words.

Reply: Done 

Line 161: typo ‘agrouped’.

Reply: Done 

Line 165: (19,23). Remove the full stop. 

Reply: Done 

Line 170-178: What is a larger space can you provide a diagram with distances i.e., 100x60 to 80x48m, competition dimension is a little vague the same for small spaces. Is it possible to define technical-tactical, activation- oriented tasks…again could you be specific player numbers would also be very useful. An overview of a typical training schedules and its contents would also help the reader. 

Reply: Done

Line 176: ‘wasn’t’ better to replace with ‘was not’ 

Reply: Done 

Line 176: ‘analized’ throughout the paper there are several different spellings of this word, please check and correct.

Reply: Done

Line 183: ‘analysed’ consistent spelling of this word is required throughout this paper.

Reply: Done

Line 192: Person’s correlation…is this reported correctly

Reply: Yes, this information is reported in the reference 24.

Line 192 to 193: Intra and inter unit reliability would be useful here to show some data. 

Reply: Thanks for your suggestion. We have included some values. 

“Intra-and inter-unit reliability was acceptable (intra-class correlation coefficient (ICC) value for the x-coordinate was 0.65 and for the y-coordinate was 0.85) for the systems analysed (25).

Line 199: This sentence may need re-wording

Reply: Done 

Line 223: Is there a reason for using the Hägglund et al paper and not the injury consensus?

Reply: We totally agree. Thanks for this point. We have changed the reference. We have included it: 

Fuller, C. W., Ekstrand, J., Junge, A., Andersen, T. E., Bahr, R., Dvorak, J., ... & Meeuwisse, W. H. (2006). Consensus statement on injury definitions and data collection procedures in studies of football (soccer) injuries. Scandinavian journal of medicine & science in sports, 16(2), 83-92.

Line 225: typo ‘te’

Reply: Done 

Line 240: Is this full training if so this would be better to clearly state.

Reply: Done. We have added this information:

Time to return to play (RTP) was calculated as the recovery time (in days) from the day of the injury until the player returned safely to training or competition. As can be seen in FC Barcelona Guide (38), the players’ demands will be increased with appropriate management of loads until he/she is ready to join 100% with the team, and when he/she performed at one total training week, from -4MD to MD, with the team we will consider the player is ready to play a game. 

Line 250: ‘We performed a descriptive study’ is this needed.

Reply: Done 

Line 262: IR could you please define what this means.

More details are required for how you calculated injury incidence with GPS metric. Again, this section is very long, but with some important information missing especially if the reader has no football background.

Reply: We have updated the table 1. We think the reader can observe how we can calculate injury incidence with GPS metric.

Results

This section is too long and needs to be concise, focus on the main findings. 

Line 316: ’34 episodes of muscle injury’ may be better to include additional information such as location and severity. The injury consensus statement could be useful here. 

Reply: We agree. We have included a table of frequency muscle injuries according to your suggestions.

Table 2: Frequency of muscle injuries 

 MUSCLE INJURY TYPE 

 “Time loss”(TL) “No time loss”(NTL) Total

CATEGORY 

Youth 11 11 22

Professional 8 4 12

TOTAL 19 19 34

POSITION 

Centre Back 2 3 5

Full back 1 2 3

Midfielders 9 8 17

Forwards 7 2 9

TOTAL 19 15 34

MATCH DAY 

-4MD 0 2 2

-3MD 6 3 9

-2MD 1 4 5

-1MD 1 3 4

 MD 9 2 12

 Pre-season 2 1 3

TOTAL 19 15 34

Discussion

Start this section with a review of the aims of the study. Then in order of importance briefly present your findings. The sub-sections of the discussion should be structured in order of importance to assist the reader and also your argument.

Reply: Thanks for this input. We have improved this section.

Line 386 to 389: The training loads are in line with the club training plan? Was this the main aim of your paper, was this the main finding?? My opinion the higher muscle injury incidence on MD-3 is the novel finding – or not? Larger spaces and players number resulted in a higher muscle injury incidence (this novel), discuss why i.e., similar physical requirements to a match day, what are these requirements and why do they increase injury risk?

Reply: Done

Our most important finding is, when we have analysed the exposure in relation to the variables of external load it is the day -3MD, which presents a higher injury incidence in all variables, HMLD (0.41x105m), HSR (1.60x105m), PL (5.06x105AU) and TD (6.61x107m) among any of the training session days surrounding the match and the match. The -3MD was where the training load was close of the competition, and the number of players was higher than match day.

Line 390 to 392: Was injury higher in match day? This is not very clear, I would re-word this sentence, also you need a reference ‘like in previous studies’ such a statement requires support.

Reply: Done

Line 393: ‘relevant differences’ I think this is not the correct wording

Reply: Thanks. We have replaced “relevant” for “remarkable”.

Line 404 to 414: The content here is very similar to that of line 169 to 174.

Reply: Thank you, we have deleted this paragraph.

Line 404 to 422: Should this be the first sub-section. Again, the structure has to reflect your major/novel findings. In my view this is one of the least important findings of your study. 

Replay: Thanks. We have reduced this paragraph and we keep this structure.

Line 411: ‘lower loads’ could you be more specific here total distance, HSR etc..

Reply: Done. We deleted this sentence.

Line 413: ‘external loads’ do you mean all variables…. please specify.

For both these points might also be worth pointing the reader to the appropriate figures to support what you are saying.

Replay: Thank you. We have deleted this paragraph.

Line 417: ‘Out with’ re-word

Reply: Done 

Line 417 to 420: Could these points be supported with a figure.

Reply: We completely agree. We have added (Figure 1 and Figure 2) at the end of the sentence. 

Line 426: ‘pro team’, you mean professional team needs to be consistent throughout the paper.

Reply: Thanks. We have corrected it.

Line 429: typos ‘stuy’ ‘they’ ‘differenecs’

Reply: Done

Line 431: Can you reference the past studies please

Replay: We didn’t find any previous studies

Line 446: Di Salvo et al. no number

Reply: Done

Line 453 to 454: Needs to be re-worded…’also’ might need to be added

Reply: Done 

Line 460 to 483: This should be the first section, if the paper is to reflect the title.

Reply: Thanks for your suggestion. As we explained before, first we provide an overview of these GPS-based measures through a descriptive analysis and subsequently we study injury incidence behavior grounded in these measures.

Line 496 and 471: Denominator is this correct word please check

Reply: Done 

Line 472 to 473: Both start with ‘We’ consider using a different word.

Reply: Done 

Line 473: ‘We really believe’ is this appropriate for a scientific journal?

Reply: Thanks for your comment. We have deleted it.

Line 476: typo ‘Then seems that’ and ‘quantyfie’, check this line.

Reply: Done

Line 490: ‘It has been shown that the injury incidence is higher when you play matches’. Replace ‘when you play matches’ with in competitive matches. Needs a reference to support your statement. 

Reply: Thanks for your important suggestion. Done

Line 490: Ekstrand 2011 no number.

Reply: Done

Line 493: ‘We..we’. Better to re-word sentence.

Reply: Done 

Line 495: ‘most important finding’…it has not been discussed until page 4 of the discussion.

We totally agree. We have moved it to the first paragraph of the Discussion section

Line 509: What do these number represent, please be clear.

Reply: Thanks. We have improved it.

Line 518: What are the three variables

Reply: We have clarified it. 

Line 520: ‘Mildfieldres’ replace with Midfielders 

Reply: Done 

Line 524: typo ‘midfielders players’ 

Reply: Done 

Line 526 to 528: If your findings agree with Bacon et al. be clear and state. What is written is unclear.

Reply: We think that it is clear. 

Line 539 to 542: How does this impact on external validity. Does this paper have an advantage of a single doctor assessing injury? If so state this and support with a reference

Reply: Thanks for your comment. Regarding external validity, we have included the following paragraph in the Limitations section:

Future studies in other soccer clubs with different training methodologies and practices must be carried out to be compared with the results of our work. “

Conclusion

The order of the conclusion is not consistent with the discussion.

Reply: Thanks. We have improved the discussion section before.

Line 551: ‘This study present that’ word missing.

Reply: Done 

Line 556: ‘will have to be involved’ needs to be re-worded.

Reply: Done 

Contributors

No mention of author AM

Reply: Done 

References:

Number 5 needs capital letters replacing with lower case in the article title.

Reply: Done

Number 11 ‘Owen2’

Reply: Done

Number 13 needs capital letters replacing with lower case in the article title.

Reply: Done

Number 17 needs capital letters replacing with lower case in the article title.

Reply: Done

Number 27 needs capital letters replacing with lower case in the article title.

Reply: Done

Number 29 please re-check

Reply: Done

Number 30 ‘H??gglund’ Wald??n’

Reply: Done

Number 37 needs capital letters replacing with lower case in the article title.

Reply: Done

---

## [Decision Letter · Decision Letter 1]

23 Feb 2021

PONE-D-20-22817R1

Descriptive analysis of GPS as exposure measures and its use in estimating the muscle injury incidence from a senior men’s and academy football team of a Spanish professional club

PLOS ONE

Dear Dr. Casals,

Thank you for submitting your manuscript to PLOS ONE. After careful consideration, we feel that it has merit but does not fully meet PLOS ONE’s publication criteria as it currently stands. Therefore, we invite you to submit a revised version of the manuscript that addresses the points raised during the review process.

We look forward to receiving your revised manuscript.

Kind regards,

Fabrizio Perroni

Academic Editor

PLOS ONE

Reviewers' comments:

Reviewer's Responses to Questions

**Comments to the Author**

1. If the authors have adequately addressed your comments raised in a previous round of review and you feel that this manuscript is now acceptable for publication, you may indicate that here to bypass the “Comments to the Author” section, enter your conflict of interest statement in the “Confidential to Editor” section, and submit your "Accept" recommendation.

Reviewer #1: (No Response)

Reviewer #2: All comments have been addressed

2. Is the manuscript technically sound, and do the data support the conclusions?

Reviewer #1: Partly

Reviewer #2: Yes

3. Has the statistical analysis been performed appropriately and rigorously? 

Reviewer #1: I Don't Know

Reviewer #2: Yes

4. Have the authors made all data underlying the findings in their manuscript fully available?

Reviewer #1: Yes

Reviewer #2: Yes

5. Is the manuscript presented in an intelligible fashion and written in standard English?

Reviewer #1: No

Reviewer #2: Yes

6. Review Comments to the Author

Reviewer #1: Thanks for the opportunity to review this paper. Again, it is clear to see that substantial effort has gone into the corrects however, there are still some issues that need to be addressed. I’m unsure if the conversation from a MS word document to a PDF is altering the format, but this does give the manuscript an untidy appearance (Line 126-129; 149-151). The entire article needs to be checked formatted correctly before submission.

Section feedback:-

Introduction

This section is too long (1011 words) and needs to be reduced by at least 50%. The main aim of this paper is ‘To investigate the external 140 load and intensities in players of a academy and professional football players according 141 to playing position and training day respect to match day’ yet there is little review of external load literature in the introduction.

Methods

Similar to the introduction the methods section is very long (1369 words) and would benefit from a significant reduction. While I empathise that the first language of authors is unlikely to be English, concise writing is required. Line 222-224 it states average physical parameters (volume and intensity) were calculated as a mean for all sessions, can you justify this method within the available literature?

Results

Again, this section is very long and could be improved by a reduction of words. To assist the reader rather than stating ‘showed very similar distributions’ please provide the data.

Discussion

Without doubt this section has been improved but the formatting is not to standard (e.g., line 415-418). However, the discussion is also very long and like the previous sections needs to reduced if it is to reach publication. More analysis of the findings are required, how does your main findings compare with the literature and if they are different why is that. To make publication there is still some work to do.

Reviewer #2: (No Response)

7. PLOS authors have the option to publish the peer review history of their article (what does this mean?). If published, this will include your full peer review and any attached files.

Reviewer #1: No

Reviewer #2: No

---

## [Author Response · Author response to Decision Letter 1]

13 Mar 2021

PLOS ONE Editorial Board 

Dear Editor,

We would like to thank you for giving us the opportunity of submitting a revision of the manuscript entitled “Descriptive analysis of GPS as exposure measures and its use in estimating the muscle injury incidence from a senior men’s and academy football team of a Spanish professional club”.

We have carefully read the comments and suggestions that have certainly improved the manuscript. Additionally, a list of changes and answers to the reviewers’ comments has been submitted.

We remain at your service for any questions during the revision process.

Sincerely,

Martí Casals

Thanks for the opportunity to review this paper. Again, it is clear to see that substantial effort has gone into the corrects however, there are still some issues that need to be addressed. I’m unsure if the conversation from a MS word document to a PDF is altering the format, but this does give the manuscript an untidy appearance (Line 126-129; 149-151). The entire article needs to be checked formatted correctly before submission.

Thank you for your comment. We have carefully read the comments and suggestions that have certainly improved the manuscript. Additionally, a list of changes and answers to the reviewers’ comments has been submitted.

Section feedback:-

Introduction

This section is too long (1011 words) and needs to be reduced by at least 50%. The main aim of this paper is ‘To investigate the external 140 load and intensities in players of a academy and professional football players according 141 to playing position and training day respect to match day’ yet there is little review of external load literature in the introduction. 

Thank you for your suggestion. We have reduced and improved the Introduction section.

Methods

Similar to the introduction the methods section is very long (1369 words) and would benefit from a significant reduction. While I empathise that the first language of authors is unlikely to be English, concise writing is required. Line 222-224 it states average physical parameters (volume and intensity) were calculated as a mean for all sessions, can you justify this method within the available literature?

Thank you for your comment that really improves the Methods section. We have tried to improve the language and we have reduced and improved the section. Moreover, we have also included available literature to justify the method calculation as you suggested. 

Results

Again, this section is very long and could be improved by a reduction of words. To assist the reader rather than stating ‘showed very similar distributions’ please provide the data. 

Thanks. We think the Results section is not too long. However, we have tried to reduce it without losing information and consider your suggestion.

Discussion

Without doubt this section has been improved but the formatting is not to standard (e.g., line 415-418). However, the discussion is also very long and like the previous sections needs to reduced if it is to reach publication. More analysis of the findings are required, how does your main findings compare with the literature and if they are different why is that.

Thank you for your suggestion. We have reduced and improved the Discussion section. Moreover, we have included this appreciation in the Conclusion subsection:

 “It is difficult for us to compare our results with other literature as, to our knowledge, this is the first study to describe the injury incidences in professional male and youth players according to GPS metrics.”

Specific Feedback

Line 68-69: Do clubs have their own economy? Better to re-write and use ‘economical consequences’

Done

Line 75: Brackets next to brackets looks untidy “(up to 5 times higher ), (2,9–11)”

Done

Line 80: Check full stop next to Thus 

Done

Line 93: Not very clear ‘if what has been planned is achieved or not’ ‘the training aim was’

Done

Line 96-101: Very long for a single sentence hard to follow.

Done

Line 106-111: Very long for a single sentence hard to follow. 

Done

Line 143: ‘day respect category’ - this is not clear

Done

Line 144: ‘the season’ – a typical season?

Done

Line 157: We

Line 159: We – consider re-wording.

Done

Line 160: This

Line 162: This – consider re-wording.

Done

Line 167: ‘Date’ – Data?

Line 167: ‘throght-out’ – typo?

Done

Line 178: Brackets next to brackets looks untidy ‘(Youth (U18 and 179 U19) and Professional)’

Line 178-185: Too long and difficult to follow.

Done

Line 187: ‘strength and power in tasks with limited spaces’ could you provide an example 

Lone 189: ‘perform tasks in larger spaces’ could you provide an example i.e., 11v11-10v10

Line 191: ‘technical-tactical elements in small spaces’ again please provide an example

Line 193: ‘activation-oriented tasks’ again please provide the reader with an example 

Thanks. We have included these suggested examples and information in a new Appendix (Appendix 1).

Line 203: was this reliability work done by your group? If not is there need to describe this, may just use the data and reference.

Done

Line 206: The

Line 208: The – consider re-wording

Line 211: The - consider re-wording 

Line 224-225: should the playing positions be capital letters?

Line 241: ‘We define an injury as 242 any injury’ – does not read clearly

Done.

Line 254: ‘eliminates interpersonal biases’ – can you support this with a reference?

Done

Line 277: Again, player positions in capitals, is this required?

Line 302: ‘a.u’ – earlier written in capitals?

Thanks. Done.

Line 316: ‘All four parameters showed very similar distributions’ - and the distributions were? Some data would be useful here?

Line 318: ‘varied according to’ – what was the variation? Data would be useful here.

Line 319: ‘pronounced for the HSR variable than for the other variables’- Data would be useful here.

Line 332: ‘similar distributions’ data would be good here

Line 333: ‘varied according to’ data would be good here

Line 336: ‘shown in Figure 1 above’ should this be here as this section relates to figure 2 – may be this needs to be moved up to figure 1 section.

Thank you for your suggestion. We have included data as you suggested in the line 319:

“pronounced for the HSR variable (median of MD: 444.41 vs median of -3MD: 227.63) than for the other variables”

Moreover, we have deleted “above” in the line 336. Thanks. 

Line 341: TL, but in Line 357 Time Loss??

Done

Line 408-411: Your points here are not very clear.

Line 438-442: So these finding (Martin-Garcia et al) are from a different club or the same club? It is not clear.

Done

Line 445-448: Not possible to discuss your findings with those in the literature? What is here provides the reader with will useful information.

Thank you so much. We have changed the wording because what we wanted to show was the reality of our club, in no case did we want to talk about what other clubs do in this regard.

Line 449-451: why could this be? Good point to discuss further – less tactical knowledge?

Thank you very much, we have referenced an article with similar findings to ours.

Line 452-456: Too many points for a single sentence.

Done

Line 463: ‘Full back’s and Forward’ capitals again…are these required?

Line 463: ‘showed slightly higher behaviour’….is this really a behaviour?

Done

Line 466: ‘and Di Salvo et al. (45) analysed the distance travelled by 300 elite players, 467 observed significant differences according to the playing position’ – this needs to be re-written.

Line 469: “Second-Strikers and Wide-Midfielders” – capitals?

Done

Line 470: How does your analysis explain the differences here could you provide more information as to why?

Line 486: ‘a total IR 2.48 (1,37 injuries training and 8,7 match)’ – is this per 1000 hours? If so please state it clearly.

Line 488: Could you include injury burden?

Line 492: ‘Nofind big differences’ – Not clear and needs better wording.

Line 484-495: ‘to know what happen’ – not very clear and would need better wording.

Line 495: ‘Then seems’ – may be a word missing here

Line 512: ‘(8.70 vs 1.37/1000 hours, P \\ .001)’ – needs a consistent approach throughout the paper.

Line 520: ‘any remarkable differences’ – what is remarkable ‘sigificant’

Done

Line 524-526: ‘Injury incidences were higher in the professional category (IRTAE = 1.59; 

IRHMLD = 0.26; IRHSR =0.95; IRLoad =3.03; IRDistance = 4.10) compared to youth players (1.34; 0.21; 0.84; 2.51; 3.3) respectively’. – would this not be explained by more compliant muscle/tendon unit, reduced force capacity etc…of younger players??

Thanks for your suggestion. The objective of the study is evaluating incidence rates of sports-related lower limb muscle injuries according to exposure time, HMLD exposure and HSR exposure taking into account the training day respect category during the season. It is a first picture in a descriptive study. We think we should do more studies in the future to try explaining it.

Line 535-536: with a notable difference – could you be clearer here what is notable significant? 

Thanks. We think it is not appropriate to add ”significant” because the aim of the study is only descriptive. 

Line 542-543: Your data agrees with Bacon et al? could this not be signposted a little better?

Done

Line 549: What 2 categories? 

Line 557: Only one club – what are the specific limitation of this?

Line 557: ‘has been’ – was? 

Done

Line 565-568: instead of list all staff members would it not be better ‘those working in academies and professional teams to better understand…’. I believe this is a really important section, but the points are not clear and do not stand out.

Line 572: ‘and midfielders’ – been in capitals all the way through the paper? 

Done

---

## [Decision Letter · Decision Letter 2]

3 Jun 2021

PONE-D-20-22817R2

Descriptive analysis of GPS as exposure measures and its use in estimating the muscle injury incidence from a senior men’s and academy football team of a Spanish professional club

PLOS ONE

Dear Dr. Casals,

Thank you for submitting your manuscript to PLOS ONE. After careful consideration, we feel that it has merit but does not fully meet PLOS ONE’s publication criteria as it currently stands. Therefore, we invite you to submit a revised version of the manuscript that addresses the points raised during the review process.

We look forward to receiving your revised manuscript.

Kind regards,

Fabrizio Perroni

Academic Editor

PLOS ONE

Reviewers' comments:

Reviewer's Responses to Questions

**Comments to the Author**

1. If the authors have adequately addressed your comments raised in a previous round of review and you feel that this manuscript is now acceptable for publication, you may indicate that here to bypass the “Comments to the Author” section, enter your conflict of interest statement in the “Confidential to Editor” section, and submit your "Accept" recommendation.

Reviewer #1: (No Response)

Reviewer #2: (No Response)

2. Is the manuscript technically sound, and do the data support the conclusions?

Reviewer #1: Partly

Reviewer #2: Partly

3. Has the statistical analysis been performed appropriately and rigorously? 

Reviewer #1: I Don't Know

Reviewer #2: No

4. Have the authors made all data underlying the findings in their manuscript fully available?

Reviewer #1: Yes

Reviewer #2: No

5. Is the manuscript presented in an intelligible fashion and written in standard English?

Reviewer #1: No

Reviewer #2: No

6. Review Comments to the Author

Reviewer #1: No doubt that this paper could make publications if the required amendments are made. However, this is the third time that I have reviewed this manuscript and the same formatting errors (i.e., single sentence paragraphs) are still present despite this being mentioned in the first draft.

Introduction

The introduction should provide a clear and targeted argument for the requirement of external load and intensities in youth and senior players according to playing position and training day by focusing on the current lack of information. While also highlighting the benefits of muscle injury incidence rates being reported using GPS metric instead of the widely accepted 1000 hours. A small section of the introduction does discuss injury incidence but this can be done in greater detail. In my view the argument within your introduction could be clearer. I have provided an example structure to the authors.

Discussion

The separate headings assist the reader which is a positive. However, there is a lack of critical analysis of the results and also comparisons with previous research could be presented better. For example, Line 358-361 is ‘list like’ and these studies presented in this section could be written more concisely “Past research has identified that defenders cover less high intensity distances compared to other outfield positions (23, 24, 25). In contrast, we found that forwards and full backs covered greater HSR distances compared to other positions. These differences might be explained by tactical roles of full back and forwards, which are unique to each clubs’ playing style. The players used in this study were encouraged by the technical staff to….. which likely resulted in greater HSR distances. Moreover, analysing match data relative to playing minutes could further explain the differences between the findings of the current study and previous work”.

Practice applications

This section is one of the most important within this manuscript , yet there is little analysis. Several of the authors are high-level applied practitioners working at the highest level of sport and it would greatly benefit this paper if more attention is paid to this section.

Introduction

Line 68-70 Consequently a focus for fitness and medical teams is to design contemporary training programmes with the objective of matching the game demands. May be use

in line with game tasks and demands to replace the words highlighted in red

Line 71-72 focusing on matching (aligning?) these game demands might paradoxically (?) also increase the risk of muscle injury during training. Can you support this statement? Matching is not a scientific word and paradoxically is there actual need for this word?

Line 72-74 Needs to be re-phrased ‘of types of drills’

Line 85-87 Unsure what the point you’re trying to say here. ‘Despite many publications in the area’ what area do you mean? ‘many publications’ you have only provided 1-reference?

Line 89-90 ‘Football injury GPS studies typically use an overall match and training injury rate’ This is unclear what you are saying here.

Methods

Line 121-125; Line 129-133; Line 186-189; Line 200-203; Line 212-214 – Are you sure these have enough content to be standalone paragraphs?

Line 135 ‘We analysed all weeks of…We analysed all training weeks

Line 137 Can you start a sentence with +1MD?

Line 139 ‘(these players performed recovery tasks)’ and these tasks are…more details needed

Line 140 ‘(these players performed compensatory load tasks)’ and these tasks are…more details needed

Line 149 ‘(Realtrack….there is no closing bracket

Line 150-151 (ICC)…can you have brackets within brackets?

Line 153 (version 927) can you have brackets within brackets?

Line 175 UEFA (17) – no full stop here…I think one is required

Line 165-179 sentence is very long and difficult to understand. Needs to be rephrased and divided into 2 or 3 separate sentences.

Line 185 ‘RTP’…this has not been defined, please define.

Line 193-194 While the content is correct, could it be rephrased and simplified so that it’s easily understood by readers with a limited background in statistics.

Line 196 ‘stratified’ would sub-group be a better phrase since it is easier to understand?

Line 216-219 Please forgive me if I am incorrect but this reads the same as line 123-125, its as if it’s been cut and pasted.

Results

Line 236-238; Line 240-244; Line 246-247; Line 255 to 256; Line 257-261 – Are you sure these have enough content to be standalone paragraphs? I think some of these can be combined or am I reading this incorrectly?

Line 257-261 This reads the same as Line 240-244. One of these sections needs to be rephrased.

Line 273-274; Line 292; Line 303; Line 313; Line 323 – brackets next to brackets…these sections should be re-formatted.

Line 281 ‘than in the’ change to compared to?

Line 287 ‘incidence by per… ‘I think you need remove ‘by’?

Discussion

Line 341-344; Line 345-349; Line 362-363; Line 383-387; Line 388-391 – again is there enough content here to create separate paragraphs?

Aim number 1….’to investigate the external load and intensities in players of a academy and professional football players according to playing position and training day respect to match day’

Aim number 2…..’to evaluate incidence rates of sports-related lower limb muscle injuries according to exposure time, HMLD exposure and HSR exposure taking into account the training respect category, positions and match day during a typical season’.

If these are the aims of this manuscript then your discussion should follow in this organisation. Line 332-337 should follow this structure.

Line 341 and Line 345 and 353 ‘In our’ a little repetitive may be change one of these.

Line 341 “in our football club’ what you write is correct but I am not sure such a statement belongs in a scientific journal.

Line 344 Great content but needs to rephrased.

Line 347-349 Needs to be rephrased

Line 355 ‘showed slightly higher values’…could you direct the reader to this data using a table or figure?

Line 355-356 ‘Our results are different from other authors’….needs rephrasing

Line 357-361 Valid points but need to be written in a concise manner that links together. At the moment they read like separate statements.

Line 390-391 You provide the same explanation twice (line 348-349), so what are the underpinning factors for this? Can you go into more details and provide some of the key characteristics of this methodology?

Limitations

Line 411-412 You correctly highlight a small sample size, single club and one season, but could you not give more information as to how these impact on your data – producing a source of bias, lacking external validity etc….

Line 414 ‘Future studies...’ I feel you could be more targeted here and give specific rather than generalising.

Line 415-416 ‘to be compared with the results of our work….’ Are you trying to say that your findings need to be validated by further research?

Reviewer #2: TITLE: Descriptive analysis of GPS as exposure measures and its use in estimating the

muscle injury incidence from a senior men’s and academy football team of a Spanish

professional club

This study investigates external load and injuries in two groups of football players from a professional football club

MAJOR COMMENTS

• There are many grammar and typing errors throughout the manuscript. The manuscript should be revised thoroughly. There are many English mistakes. In my opinion, there are mistakes from the beginning, even the title of the paper does not seem correct. A native English speaker should revise the manuscript.

• In the title and the manuscript, the authors state “senior men’s and academy football”, according to this it looks as senior players were male, but academy players were male and female? Please re-write the whole title

• There are many typing and English errors:

Line 15: Is it correctly written “Barcelonatech”? or should it be ‘BarcelonaTech’?

Line 6: Please correct the mistake: “National Institute of Physical Education fo Cataonia (INEFC) University of Barcelona, Spain”

Line 40: “ They were 30 professional male and 41 male youth academy” is not correct. Line 34 and 36: “according to categories/exposure” should be rewritten, it is not correct

Line 36-37: needs rewriting, probably it needs a comma.

Line 47-48: Should “Player load” and “Total distance” be written with capital letters as the rest of the nouns?

Line 68: “fitness (…) teams”?

Line 69: “contemporary training programs” Difficult to understand

Line 147: check and change “trainnig”

Line 335: “highes”?

Line 371: “In principle”?

[There are too many mistakes and errors for a reviewer to correct them all, please check carefully the manuscript. It is not acceptable as it is now]

ABSTRACT

• In the abstract, the authors say in line 37 that this is a descriptive epidemiological study, however I think that this study is descriptive; but it is not epidemiological. Consequently, this word should be replaced in the key words list. The same applies for line 134.

• Line 36: What do the authors mean with the term “external load exposure”?

• Line 43-46. Rewrite this sentence, it is not correct. Moreover, what does “and preseason sessions” mean? This is not clear. The authors have not explained what “sessions” means. Probably the authors could separate the sentence.

• Line 39-41: These two sentences could joined

• Line 46: eliminate “the”

• Line 52: “training session/MD” has not been specified beforehand.

• Lines 52-55: These two sentences could joined, because they repeat information. The abstract should be clear and concise.

• Line 56: The authors state “…exposure presented a similar profile”. It is difficult to understand what the authors refer to. They should clarify what are they comparing.

• Lines 58-61: On one hand, the word “presents” should be changed. On the other, it is difficult to understand the whole meaning of these sentences.

• A conclusion is lacking at the end of the abstract

• In my opinion, the word “epidemiology” should be removed from the key words as this is not an epidemiology study.

INTRODUCTION

• Lines 70-72. In correct. Perhaps remove the word “while”?

• Lines 73. What do the authors mean with the word “normal”?

• Lines 78-83. Rewrite and use commas to separate ideas.

• Line 91. Please amend this: “hours(9-12 (13,14)”

• Line 93: Please write “mesures” correctly

• Line 102: Please correct “a academy”

• Lines 105-206: This sentence should be corrected. Please change “the training respect category…season”

METHODS

• Lines 108 and 110: Remove :

• Line 110: change capital letters

• Line 118: rewrite “categorie”

• Line 118: Please rewrite: “plays the 2nd…” (“in” is lacking, perhaps)

• Information about the Ethics Committee has been explained in page 6 and 10, please delete one.

• Line 129 and 130: Replace ; by : after the teams.

• The methodology should be clearer about the registered days. Did the authors separate season and preseason training days? If so, it should be stated clearer. Moreover, they should also clarify if footballers played any matches during the preseason (any friendly matches?).

• Line 147: What does the subtitle mean? It is difficult to understand because the English is not correct

• Line 150: missing space in “Intra-and”

• Line 158: Check the verbs (have been studied?)

• Line 158: Do they only refer to the training sessions or also the match? Please clarify

• Line 159: remove the symbol [

• Line 160: THe authors should choose the format for the units and write them all in the same staile (W/kg or km·h-1 or units*min)

• Lines 158-167: Check the capital letters, it would be better to remove them all

• The methods section have improved now. Information of Table 1 should referenced (i.e. name the papers)

• Line 167. Please check the sentence: “These parameters…Table1”

• Line 169: Table 1: 5 m/s2 � this unit means that this is an acceleration, therefore it is not possible to run at a constant speed of 5 m/s2, please amend this.

• Table 1: Authors should write the abbreviations of the table

• Table 1: The authors state that the HSR is the distance covered above 21 km/h, however they do not explain the reason for that. Moreover, it is questionable that 21km/h is high speed running for all players, this would suggest that all players have the same running speed. The authors should explain the reason for choosing 21 km/h instead of 20 km/h or other velocity.

• Line 163: Why are the volume and intensity called “physical parameters”? Moreover, the authors use the term “physical intensity parameters” (line 246) which is an unusual term

• Line 174: There is a space and a . missing after (17)

• Line 177: The verbs should be in past tense

• Line 182: It is difficult to understand why the authors refer to injuries as “sports-related lower limb injuries”.

• Line 187: The authors state “…we used during the first week of the season” which is difficult to understand. Does it mean that only pre-seasonal injuries were classified according to the severity? Please clarify

• Lines 200-202. This information is redundant (see lines 158-163).

• An statistical analysis should be performed in order to compare load and injuries amongst groups (youth vs senior, positions etc). In this line, sometimes authors assume that two values are different, but because there is no real analysis it is not possible to know if the difference is significant. For example, line 281: is 1.59 larger than 1.34? How large is this difference?

RESULTS

• The quality of the figures is poor, as a consequence, it is difficult to observe the results

• Line 225: Remove “the” figure…..

• Line 231: the authors wrote “Box plots without represent outliers…” which in my opinion it is not correctly written

• Figure 1: What does “Distance” mean? Total distance? Please amend

• Line 236: Authors wrote “The global median (IQR)” however, 1) they should clarify what IQR means (interquartile range?), 2) in the statistical analysis it is not mentioned and 3) in the statistical analysis the authors state that they calculated averages, not medians. Please clarify.

• Line 237-238, 243 and 255-256: Some of the units are missing

• Line 243: gain authors mention median values, however in the statistical analysis they mention that the calculated the mean (line 195). Please clarify

• Line 246-247: Please be careful with the extra spaces between words and “min-“

• Line 255: What does “The global median” mean? Please rewrite

• Lines 242 and 257-259: Authors state that the values were higher or lower, it is necessary to apply the corresponding statistical analysis in order to ascertain if the differences are statistically sound and to quantify the differences. This analysis would reinforce the results and the power of the paper.

• Table 2: remove the title of the table form the table, they should be apart

• Table 2: Remove quotation marks

• Table 2: The number of injuries was very low, besides medical attention were included (no time loss injuries) in comparison to other studies i.e. Ekstrand J, Hägglund M, Waldén M. Epidemiology of muscle injuries in professional football (soccer). Am J Sports Med. 2011;39(6):1226-32. doi: 10.1177/0363546510395879. This issue should be discussed accordingly

• Line 281: Authors state that TL injury incidence was higher in the professional players vs youth players (1.59 vs 1.34), however, due to the fact that there is no statistical analysis this cannot be corroborated. The same applies to other results i.e. 0.25 vs 0.20 (line 289) what is the real relevance of this? Are they really “different”?. A proper statistical analysis would help to this.

DISCUSSION

• The discussion should start with an “introductory paragraph of the discussion section) including a summary paragraph and the objectives of the study.

• The discussion section is poor. Authors should discuss their results and explain them, giving reasons for it. In addition, they should compare to other studies.

• Paragraphs and sentences should follow a flow of results - ideas – comparison to other studies – discussion of the similarities/disparities….

• Practical applications: This paragraph should be rewritten in order to better clarify the real application of this study.

• Line 340: Please correct the sentence (professional and youth players?)

• Line 341-344: What is the reason for these sentences? It seems that this is an explanation for a result? Perhaps place it after the next paragraph?

• Line 345: The authors state that results of the external load was similar between professional and youth players; nevertheless, due to the low quality of the graphs it is not possible to see this results, there are no results written in the results section and there is no statistical analysis to confirm the similarities or the differences in the results.

Line 345-346: The authors explain their results but include an external reference (21). Please amend this incongruence

• Lines 353-363: check the format of the references and correct them. Check the English.

• Line 358: Difficult to understand “467 observed”

• Line 362. Check the verb tense. Also, link better both paragraphs.

• Line 368: This subtitle is not correct. Differences?

• Line 373: Correct the verb. Correct the term “what happens” it is very colloquial

• Lines 370-379: These are results; authors should avoid repeating results, unless it is necessary and discuss them, find the relevance and the meaning of the results and compare them to other authors.

• Lines 383-387: These are results again. The authors should mention the reasons for these results and the relevance

• Lines 388-391: There is an incongruence: the same training methodology leads to a difference in the injury incidence? Please explain

• Practical applications: Authors should rewrite these applications, in this version the applicability of the study is missing

• Conclusions: this paragraph is a summary of the results. Please rewrite the conclusions of the study

OVERALL COMMENTS

• The quality of the graphs is poor. It is very difficult to read the numbers and the letters

• There are many errors throughout the manuscript

• Check all the References: there are many mistakes, for example ref number 6, 11 (capital letters), 14 etc; remove [internet] and so on.

7. PLOS authors have the option to publish the peer review history of their article (what does this mean?). If published, this will include your full peer review and any attached files.

Reviewer #1: No

Reviewer #2: No

---

## [Author Response · Author response to Decision Letter 2]

12 Aug 2021

Reviewer 1

Descriptive analysis of GPS as exposure measures and its use in estimating the muscle injury incidence from a senior men’s and academy football team of a Spanish professional club

Firstly, I apologies for the slow response this was due to unforeseen circumstances.

Thanks for allowing me to review this article. Again, you can see the effort that author have ‘put in’ to developing this manuscript. Peer review is never an easy process more so when English is not your first language. I have given some points for consideration concerning the introduction and discussion. If the information is presented to a high-level this could change the way injury incidence is calculated in the literature moving forward. However, it is going to require substantial re-working. This is the third time I have reviewed this paper and the same formatting issues are still present for example… 

Thank you for your comment. We have carefully read the comments and suggestions that have certainly improved the manuscript. Additionally, a list of changes and answers to the reviewers’ comments has been submitted.

Line 121-125; Line 129-133; Line 186-189; Line 200-203; Line 212-214

Line 236-238; Line 240-244; Line 246-247; Line 255 to 256; Line 257-261

Done. Thank you

Single sentences should not form a new paragraph. Although it is possible that this is happening when converting to PDF, but checking the PDF before submission would help. 

Graphs are unclear on my laptop, but again I’m not sure of this is a issue with conversion across platforms or OS systems. 

We have followed the Figure File Requirements (https://journals.plos.org/plosone/s/figures#loc-figure-preparation-checklist) of the Plos and we haven’t had any problems when submitting it. It may be due to OS systems. However, we can ask the production team about the figures if the manuscript is finally accepted.

As I read this paper the aims are:- 

1….’to investigate the external load and intensities in players of a academy and professional football players according to playing position and training day respect to match day’

2…..’to evaluate incidence rates of sports-related lower limb muscle injuries according to exposure time, HMLD exposure and HSR exposure taking into account the training respect category, positions and match day during a typical season’.

Aim number 1 is the most important…if this is not the case then please re-structure your aims and make it clear. 

Thanks for your comment. When we chose our objectives, we decided to write them down in a chronological order of analysis, not in order of importance. That’s why, for us, the most important objective was number 2. However, after reading your comment we have rephrased this part of the section and we now have only one objective. 

The introduction should provide a clear and targeted argument for the requirement of external load and intensities in youth and senior players according to playing position and training day by focusing on the current lack of information. While also highlighting the benefits of muscle injury incidence rates being reported using GPS metric instead of the widely accepted 1000 hours. A small section of the introduction does discuss injury incidence but this can be done in greater detail. In my view the argument within your introduction could be clearer. 

Potential Introduction structure

+Game/training demands using GPS metrics

+ Few studies provide information on youth and senior players training and playing using the same training/playing methodology.

+Injury incidence rates training and games youth and senior

+How and why 1000hours lacks context

+GPS metrics go some way to providing some context; yet never used to calculate injury incidence 

Discussion

The separate headings assist the reader which is a positive. However, there is a lack of critical analysis of the results and also comparisons with previous research could be presented better. For example, Line 358-361 is ‘list like’ and these studies presented in this section could be written more concisely “Past research has identified that defenders cover less high intensity distances compared to other outfield positions (23, 24, 25). In contrast, we found that forwards and full backs covered greater HSR distances compared to other positions. These differences might be explained by tactical roles of full back and forwards, which are unique to each clubs’ playing style. The players used in this study were encouraged by the technical staff to….. which likely resulted in greater HSR distances. Moreover, analysing match data relative to playing minutes could further explain the differences between the findings of the current study and previous work”. 

Thank you for your suggestion. We have improved this section.

It is my view that every section in the discussion would benefit greatly from more critical analysis and concise writing. Of course, is always easy when (1) I speak English as a first language (2) I have hindsight on my side and likely more time to analyse what you have written. 

Thank you. A native English speaker has reviewed the manuscript.

Practice applications

This section is one of the most important within this manuscript , yet there is little analysis. Several of the authors are high-level applied practitioners working at the highest level of sport and it would greatly benefit this paper if more attention is paid to this section. 

Thank you for your suggestion. We have improved this section.

Introduction

Line 68-70 Consequently a focus for fitness and medical teams is to design contemporary training programmes with the objective of matching the game demands. May be use 

in line with game tasks and demands to replace the words highlighted in red

Line 71-72 focusing on matching (aligning?) these game demands might paradoxically (?) also increase the risk of muscle injury during training. Can you support this statement? Matching is not a scientific word and paradoxically is there actual need for this word?

Line 72-74 Needs to be re-phrased ‘of types of drills’

Line 85-87 Unsure what the point you’re trying to say here. ‘Despite many publications in the area’ what area do you mean? ‘many publications’ you have only provided 1-reference?

Line 89-90 ‘Football injury GPS studies typically use an overall match and training injury rate’ This is unclear what you are saying here.

Thank you for your comments that really improves the section. We have tried to improve the introduction section.

Methods

Line 121-125; Line 129-133; Line 186-189; Line 200-203; Line 212-214 – Are you sure these have enough content to be standalone paragraphs?

Done. Thank you 

Line 135 ‘We analysed all weeks of…We analysed all training weeks

Done. Thank you 

Line 137 Can you start a sentence with +1MD?

Done. Thank you 

Line 139 ‘(these players performed recovery tasks)’ and these tasks are…more details needed

Done. Thank you 

Line 140 ‘(these players performed compensatory load tasks)’ and these tasks are…more details needed

Done. Thank you 

Line 149 ‘(Realtrack….there is no closing bracket 

Done. Thank you 

Line 150-151 (ICC)…can you have brackets within brackets?

Done. Thank you 

Line 153 (version 927) can you have brackets within brackets?

Done. Thank you 

Line 175 UEFA (17) – no full stop here…I think one is required

Done. Thank you 

Line 165-179 sentence is very long and difficult to understand. Needs to be rephrased and divided into 2 or 3 separate sentences.

Done. Thank you 

Line 185 ‘RTP’…this has not been defined, please define.

Done. Thank you 

Line 193-194 While the content is correct, could it be rephrased and simplified so that it’s easily understood by readers with a limited background in statistics.

Done. Thank you 

Line 196 ‘stratified’ would sub-group be a better phrase since it is easier to understand?

Done. Thank you 

Line 216-219 Please forgive me if I am incorrect but this reads the same as line 123-125, its as if it’s been cut and pasted.

Done. Thank you 

Results

Line 236-238; Line 240-244; Line 246-247; Line 255 to 256; Line 257-261 – Are you sure these have enough content to be standalone paragraphs? I think some of these can be combined or am I reading this incorrectly?

Done. Thank you 

Line 257-261 This reads the same as Line 240-244. One of these sections needs to be rephrased.

Done. Thank you 

Line 273-274; Line 292; Line 303; Line 313; Line 323 – brackets next to brackets…these sections should be re-formatted.

Done. Thank you 

Line 281 ‘than in the’ change to compared to?

Done. Thank you 

Line 287 ‘incidence by per… ‘I think you need remove ‘by’?

Done. Thank you 

Discussion

Line 341-344; Line 345-349; Line 362-363; Line 383-387; Line 388-391 – again is there enough content here to create separate paragraphs? 

Done. Thank you 

Aim number 1….’to investigate the external load and intensities in players of a academy and professional football players according to playing position and training day respect to match day’

Aim number 2…..’to evaluate incidence rates of sports-related lower limb muscle injuries according to exposure time, HMLD exposure and HSR exposure taking into account the training respect category, positions and match day during a typical season’.

If these are the aims of this manuscript then your discussion should follow in this organisation. Line 332-337 should follow this structure.

Done. Thank you 

Line 341 and Line 345 and 353 ‘In our’ a little repetitive may be change one of these.

Done. Thank you 

Line 341 “in our football club’ what you write is correct but I am not sure such a statement belongs in a scientific journal.

Done. Thank you 

Line 344 Great content but needs to rephrased.

Done. Thank you 

Line 347-349 Needs to be rephrased

Done. Thank you 

Line 355 ‘showed slightly higher values’…could you direct the reader to this data using a table or figure? 

Done. Thank you 

Line 355-356 ‘Our results are different from other authors’….needs rephrasing

Done. Thank you 

Line 357-361 Valid points but need to be written in a concise manner that links together. At the moment they read like separate statements. 

Done. Thank you 

Line 390-391 You provide the same explanation twice (line 348-349), so what are the underpinning factors for this? Can you go into more details and provide some of the key characteristics of this methodology? 

Done. Thank you 

Limitations

Line 411-412 You correctly highlight a small sample size, single club and one season, but could you not give more information as to how these impact on your data – producing a source of bias, lacking external validity etc….

Line 414 ‘Future studies...’ I feel you could be more targeted here and give specific rather than generalising. 

Done. Thank you 

Line 415-416 ‘to be compared with the results of our work….’ Are you trying to say that your findings need to be validated by further research? 

Done. Thank you 

Reviewer 2:

TITLE: Descriptive analysis of GPS as exposure measures and its use in estimating the

muscle injury incidence from a senior men’s and academy football team of a Spanish

professional club

This study investigates external load and injuries in two groups of football players from a professional football club

MAJOR COMMENTS

• There are many grammar and typing errors throughout the manuscript. The manuscript should be revised thoroughly. There are many English mistakes. In my opinion, there are mistakes from the beginning, even the title of the paper does not seem correct. A native English speaker should revise the manuscript.

Done. Thank you. A native English speaker has reviewed the manuscript.

• In the title and the manuscript, the authors state “senior men’s and academy football”, according to this it looks as senior players were male, but academy players were male and female? Please re-write the whole title

Done. Thanks for your comment. We have rephrased the title. We think the new title is more accurate. 

• There are many typing and English errors: 

Line 15: Is it correctly written “Barcelonatech”? or should it be ‘BarcelonaTech’?

Done. Thank you 

Line 6: Please correct the mistake: “National Institute of Physical Education fo Cataonia (INEFC) University of Barcelona, Spain” 

Done. Thank you 

Line 40: “ They were 30 professional male and 41 male youth academy” is not correct. 

Done. Thank you 

Line 34 and 36: “according to categories/exposure” should be rewritten, it is not correct

Done. Thank you 

Line 36-37: needs rewriting, probably it needs a comma.

Done. Thank you 

Line 47-48: Should “Player load” and “Total distance” be written with capital letters as the rest of the nouns?

Done. Thank you 

Line 68: “fitness (…) teams”?

Done. Thank you 

Line 69: “contemporary training programs” Difficult to understand 

Done. Thank you 

Line 147: check and change “trainnig”

Done. Thank you 

Line 335: “highes”?

Done. Thank you 

Line 371: “In principle”?

Done. Thank you 

 [There are too many mistakes and errors for a reviewer to correct them all, please check carefully the manuscript. It is not acceptable as it is now]

ABSTRACT

• In the abstract, the authors say in line 37 that this is a descriptive epidemiological study, however I think that this study is descriptive; but it is not epidemiological. Consequently, this word should be replaced in the key words list. The same applies for line 134.

Done. Thank you 

• Line 36: What do the authors mean with the term “external load exposure”?

Done. Thank you 

• Line 43-46. Rewrite this sentence, it is not correct. Moreover, what does “and preseason sessions” mean? This is not clear. The authors have not explained what “sessions” means. Probably the authors could separate the sentence. 

Done. Thank you 

• Line 39-41: These two sentences could joined

Done. Thank you 

• Line 46: eliminate “the”

Done. Thank you 

• Line 52: “training session/MD” has not been specified beforehand. 

Done. Thank you 

• Lines 52-55: These two sentences could joined, because they repeat information. The abstract should be clear and concise.

o Line 52 we sepeak about training load

o Line 55 we sepaak about injuyri incidence.

o Must we joined?

Done. Thank you

• Line 56: The authors state “…exposure presented a similar profile”. It is difficult to understand what the authors refer to. They should clarify what are they comparing. 

Done. Thank you 

• Lines 58-61: On one hand, the word “presents” should be changed. On the other, it is difficult to understand the whole meaning of these sentences.

Done. Thank you 

• A conclusion is lacking at the end of the abstract

Done. Thank you 

• In my opinion, the word “epidemiology” should be removed from the key words as this is not an epidemiology study. 

Done. Thank you 

INTRODUCTION

• Lines 70-72. In correct. Perhaps remove the word “while”?

Done. Thank you 

• Lines 73. What do the authors mean with the word “normal”?

Done. Thank you 

• Lines 78-83. Rewrite and use commas to separate ideas.

Done. Thank you 

• Line 91. Please amend this: “hours(9-12 (13,14)”

Done. Thank you 

• Line 93: Please write “mesures” correctly

Done. Thank you 

• Line 102: Please correct “a academy”

Done. Thank you 

• Lines 105-206: This sentence should be corrected. Please change “the training respect category…season”

Done. Thank you 

METHODS

• Lines 108 and 110: Remove :

Done. Thank you 

• Line 110: change capital letters

Done. Thank you 

• Line 118: rewrite “categorie”

Done. Thank you 

• Line 118: Please rewrite: “plays the 2nd…” (“in” is lacking, perhaps)

Done. Thank you 

• Information about the Ethics Committee has been explained in page 6 and 10, please delete one. 

Done. Thank you 

• Line 129 and 130: Replace ; by : after the teams. 

Done. Thank you 

• The methodology should be clearer about the registered days. Did the authors separate season and preseason training days? If so, it should be stated clearer. Moreover, they should also clarify if footballers played any matches during the preseason (any friendly matches?). 

Done. Thank you 

• Line 147: What does the subtitle mean? It is difficult to understand because the English is not correct

Done. Thank you 

• Line 150: missing space in “Intra-and”

Done. Thank you 

• Line 158: Check the verbs (have been studied?)

Done. Thank you 

• Line 158: Do they only refer to the training sessions or also the match? Please clarify

Done. Thank you 

• Line 159: remove the symbol [

Done. Thank you 

• Line 160: THe authors should choose the format for the units and write them all in the same staile (W/kg or km·h-1 or units*min)

Done. Thank you 

• Lines 158-167: Check the capital letters, it would be better to remove them all

Done. Thank you 

• The methods section have improved now. Information of Table 1 should referenced (i.e. name the papers)

Done. Thank you 

• Line 167. Please check the sentence: “These parameters…Table1”

Done. Thank you 

• Line 169: Table 1: 5 m/s2 � this unit means that this is an acceleration, therefore it is not possible to run at a constant speed of 5 m/s2, please amend this. 

Done. Thank you 

• Table 1: Authors should write the abbreviations of the table

Done. Thank you 

• Table 1: The authors state that the HSR is the distance covered above 21 km/h, however they do not explain the reason for that. Moreover, it is questionable that 21km/h is high speed running for all players, this would suggest that all players have the same running speed. The authors should explain the reason for choosing 21 km/h instead of 20 km/h or other velocity.

Thank you very much for your contribution. The registration of previous sessions with other GPS system is a decision of the club.

• Line 163: Why are the volume and intensity called “physical parameters”? Moreover, the authors use the term “physical intensity parameters” (line 246) which is an unusual term.

Done. Thank you 

• Line 174: There is a space and a . missing after (17)

Done. Thank you 

• Line 177: The verbs should be in past tense

Done. Thank you 

• Line 182: It is difficult to understand why the authors refer to injuries as “sports-related lower limb injuries”. 

Done. Thank you 

• Line 187: The authors state “…we used during the first week of the season” which is difficult to understand. Does it mean that only pre-seasonal injuries were classified according to the severity? Please clarify

Done. Thank you 

• Lines 200-202. This information is redundant (see lines 158-163).

• An statistical analysis should be performed in order to compare load and injuries amongst groups (youth vs senior, positions etc). In this line, sometimes authors assume that two values are different, but because there is no real analysis it is not possible to know if the difference is significant. For example, line 281: is 1.59 larger than 1.34? How large is this difference?

Thanks. The purpose of our article is merely descriptive. That is, we are just describing our sample doing comparisons between some observed indicators. As you point out, if we would want to do inference about these comparisons, we should perform some statistical analysis taking into account the intra-individual correlation for each player. Furthermore, the external validation of the inference made to players from other teams would be compromised by having a sample of players from the same club. Therefore, our aim is to show a first descriptive picture of some relevant indicators. 

RESULTS

• The quality of the figures is poor, as a consequence, it is difficult to observe the results

Done. Thank you 

• Line 225: Remove “the” figure…..

Done. Thank you 

• Line 231: the authors wrote “Box plots without represent outliers…” which in my opinion it is not correctly written

Done. Thank you 

• Figure 1: What does “Distance” mean? Total distance? Please amend

Done. Thank you 

• Line 236: Authors wrote “The global median (IQR)” however, 1) they should clarify what IQR means (interquartile range?), 2) in the statistical analysis it is not mentioned and 3) in the statistical analysis the authors state that they calculated averages, not medians. Please clarify. 

Thank you for your suggestion. The Box-plots show the median and the interquartile range (IQR) of the averages of each player along all the season. We have included and clarified it as you suggested in the Statistical Analysis section.

• Line 237-238, 243 and 255-256: Some of the units are missing

Done. Thank you 

• Line 243: gain authors mention median values, however in the statistical analysis they mention that the calculated the mean (line 195). Please clarify

Thanks. As it was written before, we have included and clarified it as you suggested in the Statistical Analysis section. First, we calculated the mean in the process of wrangling and get the data. Then, we described the data we used the median and the interquartile range (IQR).

• Line 246-247: Please be careful with the extra spaces between words and “min-“

Done. Thank you 

• Line 255: What does “The global median” mean? Please rewrite

Done. Thank you 

• Lines 242 and 257-259: Authors state that the values were higher or lower, it is necessary to apply the corresponding statistical analysis in order to ascertain if the differences are statistically sound and to quantify the differences. This analysis would reinforce the results and the power of the paper.

Thanks for your suggestion. In the Figure 1 (box plots) we only describe the pattern of the sample. From the point of view of Miguel Hernan or Rasmus Nielsen, we have described the aim of our study as just descriptive and does not intend to use inference.

Hernán, M. A., Hsu, J., & Healy, B. (2019). A second chance to get causal inference right: a classification of data science tasks. Chance, 32(1), 42-49.

Nielsen, R. O., Simonsen, N. S., Casals, M., Stamatakis, E., & Mansournia, M. A. (2020). Methods matter and the ‘too much, too soon’theory (part 2): what is the goal of your sports injury research? Are you describing, predicting or drawing a causal inference?.

• Table 2: remove the title of the table form the table, they should be apart

Done. Thank you 

• Table 2: Remove quotation marks

Done. Thank you 

• Table 2: The number of injuries was very low, besides medical attention were included (no time loss injuries) in comparison to other studies i.e. Ekstrand J, Hägglund M, Waldén M. Epidemiology of muscle injuries in professional football (soccer). Am J Sports Med. 2011;39(6):1226-32. doi: 10.1177/0363546510395879. This issue should be discussed accordingly

• Line 281: Authors state that TL injury incidence was higher in the professional players vs youth players (1.59 vs 1.34), however, due to the fact that there is no statistical analysis this cannot be corroborated. The same applies to other results i.e. 0.25 vs 0.20 (line 289) what is the real relevance of this? Are they really “different”?. A proper statistical analysis would help to this. 

As pointed above, we have described the pattern of the sample.

DISCUSSION

• The discussion should start with an “introductory paragraph of the discussion section) including a summary paragraph and the objectives of the study.

• The discussion section is poor. Authors should discuss their results and explain them, giving reasons for it. In addition, they should compare to other studies. 

• Paragraphs and sentences should follow a flow of results - ideas – comparison to other studies – discussion of the similarities/disparities….

• Practical applications: This paragraph should be rewritten in order to better clarify the real application of this study. 

• Line 340: Please correct the sentence (professional and youth players?)

Done. Thank you

• Line 341-344: What is the reason for these sentences? It seems that this is an explanation for a result? Perhaps place it after the next paragraph?

Done. Thank you

• Line 345: The authors state that results of the external load was similar between professional and youth players; nevertheless, due to the low quality of the graphs it is not possible to see this results, there are no results written in the results section and there is no statistical analysis to confirm the similarities or the differences in the results. 

We have followed the Figure File Requirements (https://journals.plos.org/plosone/s/figures#loc-figure-preparation-checklist) of the Plos and we haven’t had any problems when submitting it. It may be due to OS systems. 

The description of Figures 1 and 2 are not too long and we only report the general pattern because the main objective of the study is to explain the incidences. 

• Line 345-346: The authors explain their results but include an external reference (21). Please amend this incongruence

Done. Thank you

• Lines 353-363: check the format of the references and correct them. Check the English.

Done. Thank you

• Line 358: Difficult to understand “467 observed”

Done. Thank you

• Line 362. Check the verb tense. Also, link better both paragraphs.

Done. Thank you

• Line 368: This subtitle is not correct. Differences? 

Done. Thank you

• Line 373: Correct the verb. Correct the term “what happens” it is very colloquial 

Done. Thank you

• Lines 370-379: These are results; authors should avoid repeating results, unless it is necessary and discuss them, find the relevance and the meaning of the results and compare them to other authors. 

To our knowledge, it is the first time that such analysis is performed in elite footballers. 

• Lines 383-387: These are results again. The authors should mention the reasons for these results and the relevance

Done. Thank you

• Lines 388-391: There is an incongruence: the same training methodology leads to a difference in the injury incidence? Please explain

Done. Thank you

• Practical applications: Authors should rewrite these applications, in this version the applicability of the study is missing

• Conclusions: this paragraph is a summary of the results. Please rewrite the conclusions of the study 

OVERALL COMMENTS 

• The quality of the graphs is poor. It is very difficult to read the numbers and the letters

• There are many errors throughout the manuscript

• Check all the References: there are many mistakes, for example ref number 6, 11 (capital letters), 14 etc; remove [internet] and so on.

Done. Thank you

---

## [Decision Letter · Decision Letter 3]

21 Jan 2022

Use of GPS to measure external load and estimate the incidence of muscle injuries in men’s football: a novel descriptive study

PONE-D-20-22817R3

Dear Dr. Casals,

We’re pleased to inform you that your manuscript has been judged scientifically suitable for publication and will be formally accepted for publication once it meets all outstanding technical requirements.

Kind regards,

Fabrizio Perroni

Academic Editor

PLOS ONE

Additional Editor Comments (optional):

Reviewers' comments:

Reviewer's Responses to Questions

**Comments to the Author**

1. If the authors have adequately addressed your comments raised in a previous round of review and you feel that this manuscript is now acceptable for publication, you may indicate that here to bypass the “Comments to the Author” section, enter your conflict of interest statement in the “Confidential to Editor” section, and submit your "Accept" recommendation.

Reviewer #2: All comments have been addressed

2. Is the manuscript technically sound, and do the data support the conclusions?

Reviewer #2: Yes

3. Has the statistical analysis been performed appropriately and rigorously? 

Reviewer #2: Yes

4. Have the authors made all data underlying the findings in their manuscript fully available?

Reviewer #2: Yes

5. Is the manuscript presented in an intelligible fashion and written in standard English?

Reviewer #2: Yes

6. Review Comments to the Author

Reviewer #2: Thanks to the authors. All the comments have been responded in my opinion the manuscript has improved now and it is ready for publication

7. PLOS authors have the option to publish the peer review history of their article (what does this mean?). If published, this will include your full peer review and any attached files.

Reviewer #2: No

---

## [Editor Report · Acceptance letter]

26 Jan 2022

PONE-D-20-22817R3 

Use of GPS to measure external load and estimate the incidence of muscle injuries in men’s football: a novel descriptive study 

Dear Dr. Casals:

I'm pleased to inform you that your manuscript has been deemed suitable for publication in PLOS ONE. Congratulations! Your manuscript is now with our production department. 

Kind regards, 

on behalf of

Dr. Fabrizio Perroni 

Academic Editor

PLOS ONE